# High lumenal chloride in the lysosome is critical for lysosome function

**Kasturi Chakraborty[1,2], KaHo Leung[1,2], Yamuna Krishnan[1,2]***

[1]Department of Chemistry, University of Chicago, Chicago, United States; [2]Grossman Institute of Neuroscience, Quantitative Biology and Human Behavior, University of Chicago, Chicago, United States

**Abstract** Lysosomes are organelles responsible for the breakdown and recycling of cellular machinery. Dysfunctional lysosomes give rise to lysosomal storage disorders as well as common neurodegenerative diseases. Here, we use a DNA-based, fluorescent chloride reporter to measure lysosomal chloride in *Caenorhabditis elegans* as well as murine and human cell culture models of lysosomal diseases. We find that the lysosome is highly enriched in chloride, and that chloride reduction correlates directly with a loss in the degradative function of the lysosome. In nematodes and mammalian cell culture models of diverse lysosomal disorders, where previously only lysosomal pH dysregulation has been described, massive reduction of lumenal chloride is observed that is $\sim 10^3$ fold greater than the accompanying pH change. Reducing chloride within the lysosome impacts $Ca^{2+}$ release from the lysosome and impedes the activity of specific lysosomal enzymes indicating a broader role for chloride in lysosomal function.

## Introduction

Chloride is the most abundant, soluble anion in the body. Cytosolic chloride can be as low as ~45 mM, while extracellular chloride is ~110 mM (*Treharne et al., 2006*), (*Sonawane et al., 2002*). Chloride concentration values thus span a wide range and yet, in each compartment, it is quite tightly regulated (*Sonawane and Verkman, 2003*). For example, in early endosomes it is ~40 mM, late endosomes it is ~70 mM and lysosomes it is ~108 mM (*Hara-Chikuma et al., 2005*; *Saha et al., 2015*; *Sonawane et al., 2002*). Chloride levels are stringently regulated by chloride channels such as cystic fibrosis transmembrane regulator (CFTR), the CLC family of channels or calcium activated chloride channels, and their dysregulation is directly linked to several diseases including cystic fibrosis, myotonia, epilepsy, hyperekplexia or deafness (*Planells-Cases and Jentsch, 2009*). Chloride is largely considered to function as a counter ion only to balance changes in cation fluxes related to signaling (*Scott and Gruenberg, 2011*). In one form, this balancing function serves to reset the membrane potential of depolarized neurons through the operation of plasma membrane resident chloride channels/exchangers (*Chen, 2005*). In another form, it serves to continuously facilitate organelle acidification, through the operation of intracellular chloride channels (*Stauber and Jentsch, 2013*). Despite its importance in cell function, intracellular chloride has never been visualized or quantitated *in vivo*.

DNA nanotechnology has offered creative, functional imaging solutions to quantitate second messengers as well as image organelles in real time in living cells and in genetic model organisms (*Bhatia et al., 2016*; *Chakraborty et al., 2016*; *Krishnan and Bathe, 2012*; *Surana et al., 2015*). Here, using a previously developed, pH-independent, DNA-based fluorescent chloride reporter called *Clensor*, we have made the first measure of chloride in a live multicellular organism, creating *in vivo* chloride maps of lysosomes in *C. elegans*.

***For correspondence:** yamuna@uchicago.edu

**Competing interests:** The authors declare that no competing interests exist.

**eLife digest** In cells, worn out proteins and other unnecessary materials are sent to small compartments called lysosomes to be broken down and recycled. Lysosomes contain many different proteins including some that break down waste material into recyclable fragments and others that transport the fragments out of the lysosome. If any of these proteins do not work, waste products build up and cause disease. There are around 70 such lysosomal storage diseases, each arising from a different lysosomal protein not working correctly.

A recently developed "nanodevice" called *Clensor* can measure the levels of chloride ions inside cells. *Clensor* is constructed from DNA, and its fluorescence changes when it detects chloride ions. Although chloride ions have many biological roles, chloride ion levels had not been measured inside a living organism. Now, Chakraborty et al. – including some of the researchers who developed Clensor – have used this nanodevice to examine chloride ion levels in the lysosomes of the roundworm *Caenorhabditis elegans*. This revealed that the lysosomes contain high levels of chloride ions. Furthermore, reducing the amount of chloride in the lysosomes made them worse at breaking down waste.

Do lysosomes affected by lysosome storage diseases also contain low levels of chloride ions? To find out, Chakraborty et al. used *Clensor* to study *C. elegans* worms and mouse and human cells whose lysosomes accumulate waste products. In all these cases, the levels of chloride in the diseased lysosomes were much lower than normal. This had a number of effects on how the lysosomes worked, such as reducing the activity of key lysosomal proteins.

Chakraborty et al. also found that *Clensor* can be used to distinguish between different lysosomal storage diseases. This means that in the future, *Clensor* (or similar methods that directly measure chloride ion levels in lysosomes) may be useful not just for research purposes. They may also be valuable for diagnosing lysosomal storage diseases early in infancy that, if left undiagnosed, are fatal.

Our investigations reveal that lysosomal chloride levels *in vivo* are even higher than extracellular chloride levels. Others and we have shown that lysosomes have the highest lumenal acidity and the highest lumenal chloride , among all endocytic organelles (*Saha et al., 2015*; *Weinert et al., 2010*). Although lumenal acidity has been shown to be critical to the degradative function of the lysosome (*Appelqvist et al., 2013*; *Eskelinen et al., 2003*), the necessity for such high lysosomal chloride is unknown. In fact, in many lysosomal storage disorders, lumenal hypoacidification compromises the degradative function of the lysosome leading to the toxic build-up of cellular cargo targeted to the lysosome for removal, resulting in lethality (*Guha et al., 2014*). Lysosomal storage disorders (LSDs) are a diverse collection of ~70 different rare, genetic diseases that arise due to dysfunctional lyso-somes (*Samie and Xu, 2014*). Dysfunction in turn arises from mutations that compromise protein transport into the lysosome, the function of lysosomal enzymes, or lysosomal membrane integrity (*Futerman and van Meer, 2004*). Importantly, for a sub-set of lysosomal disorders like osteopetrosis or neuronal ceroid lipofuscinoses (NCL), lysosomal hypoacidification is not observed (*Kasper et al., 2005*). Both these conditions result from a loss of function of the lysosomal H$^+$-Cl$^-$ exchange trans-porter CLC-7 (*Kasper et al., 2005*). In both mice and flies, lysosomal pH is normal, yet both mice and flies were badly affected (*Poët et al., 2006*; *Weinert et al., 2010*).

The lysosome performs multiple functions due to its highly fusogenic nature. It fuses with the plasma membrane to bring about plasma membrane repair as well as lysosomal exocytosis, it fuses with the autophagosome to bring about autophagy, it is involved in nutrient sensing and it fuses with endocytic cargo to bring about cargo degradation (*Appelqvist et al., 2013*; *Xu and Ren, 2015*). To understand which, if any, of these functions is affected by chloride dysregulation, we chose to study genes related to osteopetrosis in the versatile genetic model organism *Caenorhabditis elegans*. By leveraging the DNA scaffold of *Clensor* as a natural substrate along with its ability to quantitate chloride, we could simultaneously probe the degradative capacity of the lysosome *in vivo* and then in cultured mammalian cells. Our findings reveal that depleting lysosomal chloride showed a direct correlation with loss of the degradative function of the lysosome. We found that lowering

lysosomal chloride also reduced the level of $Ca^{2+}$ released from the lysosome. We also observed that reduction of lysosomal chloride inhibited the activity of specific lysosomal enzymes such as cathepsin C and arylsulfatase B. The role of chloride in defective lysosomal degradation has been hypothesized in the past (*Stauber and Jentsch, 2013*; *Wartosch and Stauber, 2010*; *Wartosch et al., 2009*), and our studies provide the first mechanistic proof of a broader role for chloride in lysosome function.

## Results and discussion

### Reporter design and uptake pathway in coelomocytes of *C. elegans*

In this study we use two DNA nanodevices, called the I-switch and *Clensor*, to fluorescently quantitate pH and chloride respectively (*Modi et al., 2009*; *Saha et al., 2015*). The I-switch is composed of two DNA oligonucleotides. One of these can form an i-motif, which is an unusual DNA structure formed by protonated cytosines (*Gehring et al., 1993*). In the I-switch, intrastrand i-motif formation is used to bring about a pH-dependent conformational change, that leverages fluorescence resonance energy transfer (FRET) to create a ratiometric fluorescent pH reporter. (*Figure 1—figure supplement 2*)

The DNA-based chloride sensor, *Clensor,* is composed of three modules: a sensing module, a normalizing module and a targeting module (*Figure 1a*) (*Saha et al., 2015*; *Prakash et al., 2016*). The sensing module is a 12 base long peptide nucleic acid (PNA) oligomer conjugated to a fluorescent, chloride-sensitive molecule 10,10′-Bis[3-carboxypropyl]−9,9′-biacridinium dinitrate (BAC), (*Figure 1a*) (*Sonawane et al., 2002*). The normalizing module is a 38 nt DNA sequence bearing an Alexa 647 fluorophore that is insensitive to Cl. The targeting module is a 26 nt double stranded DNA domain that targets it to the lysosome via the endolysosomal pathway by engaging the scavenger receptor or ALBR pathway. In physiological environments, BAC specifically undergoes collisional quenching by Cl, thus lowering its fluorescence intensity (G) linearly with increasing Cl concentrations. In contrast, the fluorescence intensity of Alexa 647 (R) remains constant (*Figure 1b*). This results in R/G ratios of *Clensor* emission intensities varying linearly with [Cl] over the entire physiological regime of [Cl]. Since the response of *Clensor* is insensitive to pH changes, it enables the quantitation of lumenal chloride in organelles of living cells regardless of their lumenal pH (*Saha et al., 2015*).

### Targeting *Clensor* to lysosomes of coelomocytes in *C. elegans*

Coelomocytes of *C. elegans* are known to endocytose foreign substances injected in the body cavity (*Fares and Greenwald, 2001*). The polyanionic phosphate backbone of DNA can be co-opted to target it to scavenger receptors and thereby label organelles on the endolysosomal pathway in tissue macrophages and coelomocytes in *C. elegans* (*Figure 1c and d*) (*Bhatia et al., 2011*; *Modi et al., 2009*; *Saha et al., 2015*; *Surana et al., 2011*). Alexa 647 labelled I-switch (I4$^{cLY}$) and *Clensor* were each injected in the pseudocoelom of 1-day-old adult worms expressing *pmyo-3*::ssGFP. In these worms, soluble GFP synthesized in muscles and secreted into the pseudocoelom is actively internalized by the coelomocytes resulting in GFP labeling of the coelomocytes (*Fares and Greenwald, 2001*). After 1 hr, both devices quantitatively colocalize with GFP indicating that they specifically mark endosomes in coelomocytes (*Figure 1e* and *Figure 1—figure supplement 1c*). Endocytic uptake of DNA nanodevices was performed in the presence of 30 equivalents of maleylated bovine serum albumin (mBSA), a well-known competitor for the anionic ligand binding receptor (ALBR) pathway (*Gough and Gordon, 2000*). Coelomocyte labeling by I4$^{cLY}$or *Clensor* were both efficiently competed out by mBSA indicating that both reporters were internalized by ALBRs and trafficked along the endolysosomal pathway (*Figure 1—figure supplement 1b*) (*Surana et al., 2011*).

### *In vivo* performance of DNA reporters

Next, the functionality of I4$^{cLY}$ and *Clensor* were assessed *in vivo*. To generate an *in vivo* calibration curve for the I-switch I4$^{cLY}$, coelomocytes labeled with I4$^{cLY}$ were clamped at various pH values between pH 4 and 7.5 as described previously and in the supporting information (*Surana et al., 2011*). This indicated that, as expected, the I-switch showed *in vitro* and *in vivo* performance

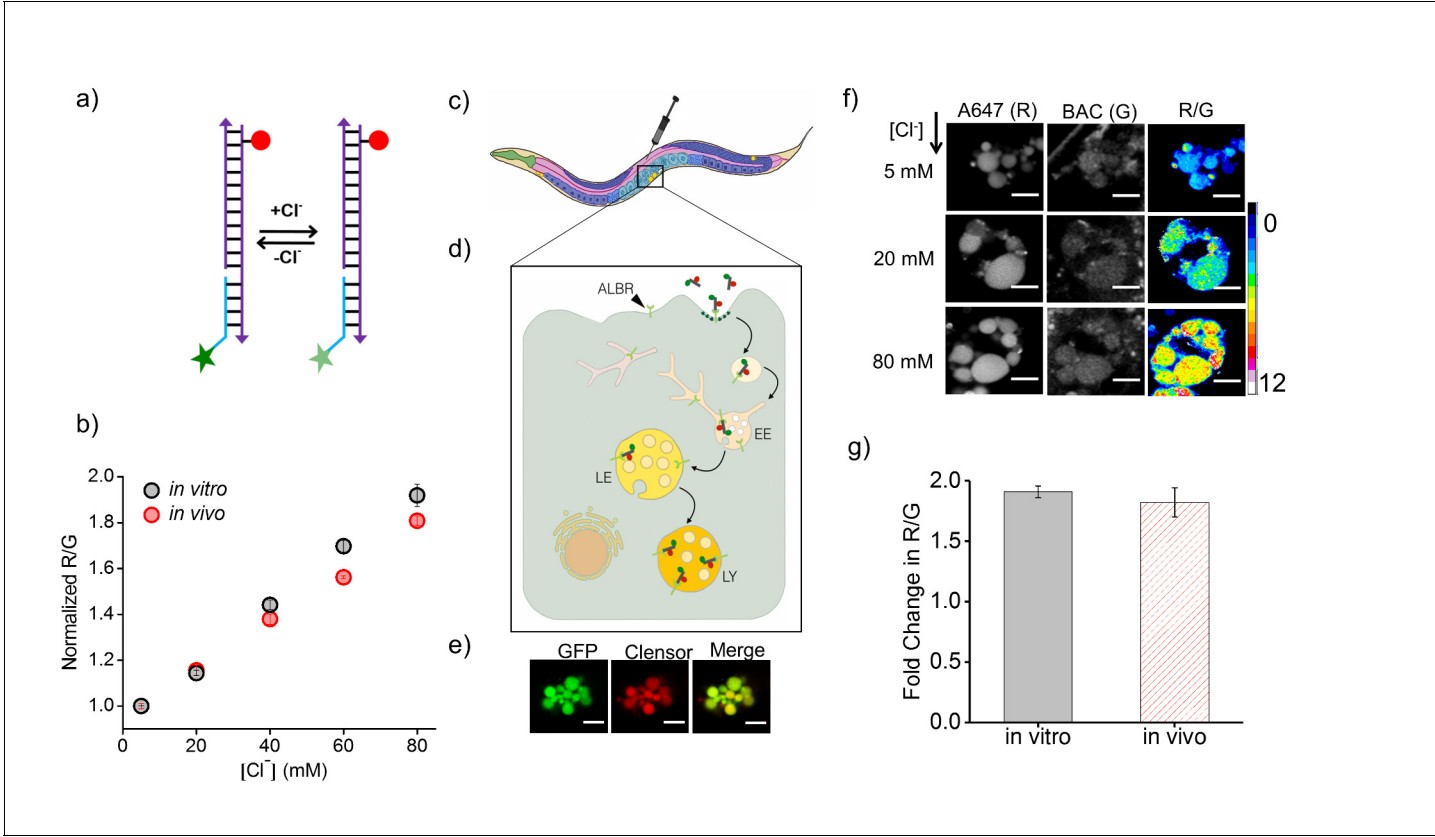

**Figure 1.** *Clensor* recapitulates its chloride sensing characteristics *in vivo*. (**a**) Schematic of the ratiometric, fluorescent chloride (Cl⁻) reporter *Clensor*. It bears a Cl⁻ sensitive fluorophore, BAC (green star) and a Cl⁻ insensitive fluorophore, Alexa 647 (red circle) (**b**) Calibration profile of *Clensor in vitro* (grey) and *in vivo* (red) given by normalized Alexa 647 (**R**) and BAC (**G**) intensity ratios versus [Cl⁻]. (**c**) Receptor mediated endocytic uptake of *Clensor* in coelomocytes post injection in *C. elegans*. (**d**) *Clensor* is trafficked by the anionic ligand binding receptor (ALBR) from the early endosome (EE) to the late endosome (LE) and then lysosome (LY). (**e**) Colocalization of *Clensor*$_{A647}$ (red channel) microinjected in the pseudocoelom with GFP-labeled coelomocytes (green channel). Scale bar: 5 µm. (**f**) Representative fluorescence images of endosomes in coelomocytes labeled with *Clensor* and clamped at the indicated Cl⁻ concentrations ([Cl⁻]). Images are acquired in the Alexa 647 (**R**) and BAC (**G**) channels from which corresponding pseudocolored R/G images are generated. The *in vivo* calibration profile is shown in (**b**). Scale bar: 5 µm. Error bars indicate s.e.m. (n = 15 cells,≥50 endosomes) (**g**) *In vitro* (grey) and *in vivo* (red) fold change in R/G ratios of *Clensor* from 5 mM to 80 mM [Cl⁻].

The following figure supplements are available for figure 1:

**Figure supplement 1.** (**a**) Quantification of co-localization between DNA nanodevices and GFP in *arIs37* worms.

**Figure supplement 2.** (**a**) Schematic of a DNA nanodevice, *I-switch*, that functions as a fluorescent pH reporter based on a pH triggered conformational change that is transduced to photonic changes driven by differential fluorescent resonance energy transfer between donor (D, green) and acceptor (A, red) fluorophores (**b**) pH calibration curve of I4$^{cLY}_{A488/A647}$ *in vivo* (red) and *in vitro* (grey) showing normalized D/A ratios versus pH.

**Figure supplement 3.** Selectivity of *Clensor* (200 nM) in terms of its fold change in R/G from ~0 to 100 mM of each indicated anion unless otherwise indicated.

characteristics that were extremely well matched (*Figure 1—figure supplement 2b–e*). To assess the *in vivo* functionality of *Clensor*, a standard Cl⁻ calibration profile was generated by clamping the lumenal [Cl⁻] to that of an externally added buffer containing known [Cl⁻] as described previously for cultured cells (*Saha et al., 2015*). Endosomes of coelomocytes were labeled with *Clensor* and fluorescence images were acquired in the BAC channel (G) as well as Alexa 647 channel (R) as described (see Materials and methods), from which were obtained R/G ratios of every endosome clamped at a specific [Cl⁻] (*Figure 1b*). Endosomal R/G ratios showed a linear dependence on [Cl⁻] with ~2 fold

change in R/G values from 5 mM to 80 mM [Cl] (*Figure 1f and g*). This is very well matched with its *in vitro* fold change in R/G over the same regime of [Cl].

## DNA nanodevices localize specifically in lysosomes in diverse genetic backgrounds

Before performing quantitative chloride imaging in various mutant nematodes, we checked whether lysosomal targeting of *Clensor* and the I-switch were preserved in a variety of genetic backgrounds of our interest. *Clensor* was injected into LMP-1::GFP worms treated with RNAi against specific lysosomal storage disorder (LSD)-related genes or genes linked to osteopetrosis. We observed significant colocalization (>74%) of *Clensor* with LMP-1-GFP labeled lysosomes in these coelomocytes (*Figure 3—figure supplement 2b and c*). Given that both *Clensor* and the I-switch robustly labeled lysosomes of coelomocytes in wild type worms (N2), mutants and RNAi knockdowns of a range of LSD-related genes, we explored whether these devices could report on alterations, if any, in the lumenal ionicity in these lysosomes, and thereby possibly report on lysosome dysfunction.

## Quantitative *in vivo* imaging of chloride in lysosomes in *C. elegans*

As an initial study, we focused on *C. elegans* nematodes in which genes related to osteopetrosis are mutated. Osteopetrosis results from non-functional osteoclasts that lead to increased bone mass and density due to a failure in bone resorption (*Sobacchi et al., 2013*). In humans, osteopetrosis results from mutations in a lysosomal chloride-proton antiporter CLCN7, and its auxiliary factor OSTM1 (*Figure 2a*) (*Kornak et al., 2001*; *Lange et al., 2006*). It also results from mutations in TCIRG1, which is the a3 subunit of a lysosomal V-ATPase (*Kornak et al., 2000*) and SNX-10, a sorting nexin implicated in lysosome transport to form the ruffled border of osteoclasts, which is critical for osteoclast function (*Aker et al., 2012*) (*Figure 2a*). Lysosomes of CLC7 knockout mice show normal lumenal pH, yet the mice manifest osteopetrosis as well as neurodegeneration, indicating that despite the apparently normal lumenal milieu, the organelle is still dysfunctional (*Kasper et al., 2005*). The *C. elegans* homologs for these genes are *clh-6* (CLCN7), *F42A8.3* (*ostm-1*; OSTM1), *unc-32* (TCIRG1) and *snx-3* (SNX10) (*Figure 2a*).

Clensor was injected into N2, *clh-6* and *unc-32* mutants and RNAi knockdowns of *ostm-1* and *snx-3*. Chloride concentrations in the lysosomes of each genetic background at 60 min post injection were obtained (*Figure 2b,c* and *Figure 2—figure supplement 1*). N2 worms showed a chloride concentration of ~75 mM. Knocking down *clh-6* and *ostm-1* resulted in a dramatic decrease of lysosomal chloride to ~45 mM due to the loss of function in chloride transport. Lumenal pH in the lysosomes of these mutants was normal, consistent with findings in both flies and mice (*Saha et al., 2015*; *Weinert et al., 2010*). As a control, knocking down a plasma membrane resident CLC channel such as *clh-4* showed no effect on either lysosomal chloride or pH (*Schriever et al., 1999*). *unc-32c* is a non-functional mutant of the V-ATPase *a* sub-unit, while *unc-32f* is a hypomorph (*Pujol et al., 2001*). Interestingly, a clear inverse correlation with *unc-32* functionality was obtained when comparing their lysosomal chloride levels i.e.,~55 mM and ~65 mM for *unc-32c* and *unc-32f* respectively. Importantly, *snx-3* knockdowns showed lysosomal chloride levels that mirrored those of wild type lysosomes. In all genetic backgrounds, we observed that lysosomal chloride concentrations showed no correlation with lysosome morphology (*Figure 3—figure supplement 1d*).

## Reducing lumenal chloride lowers the degradative capacity of the lysosome

Dead and necrotic bone cells release their endogenous chromatin extracellularly - thus duplex DNA constitutes cellular debris and is physiologically relevant cargo for degradation in the lysosome of phagocytic cells (*Elmore, 2007*; *Luo and Loison, 2008*). Coelomocytes are phagocytic cells of *C. elegans*, and thus, the half-life of *Clensor* or I4$^{cLY}$ in these cells constitutes a direct measure of the degradative capacity of the lysosome (*Tahseen, 2009*). We used a previously established assay to measure the half-life of I-switches in lysosomes (*Surana et al., 2013*). Worms were injected with 500 nM I4$^{cLY}$ and the fluorescence intensity obtained in 10 cells at each indicated time point was quantitated as a function of time. The I-switch I4$^{cLY}$ had a half-life of ~6 hr in normal lysosomes, which nearly doubled when either *clh-6* or *ostm-1* were knocked down (*Figure 2d* and *Figure 2—figure supplement 2*). Both *unc-32c* and *unc-32f* mutants showed near-normal lysosome degradation

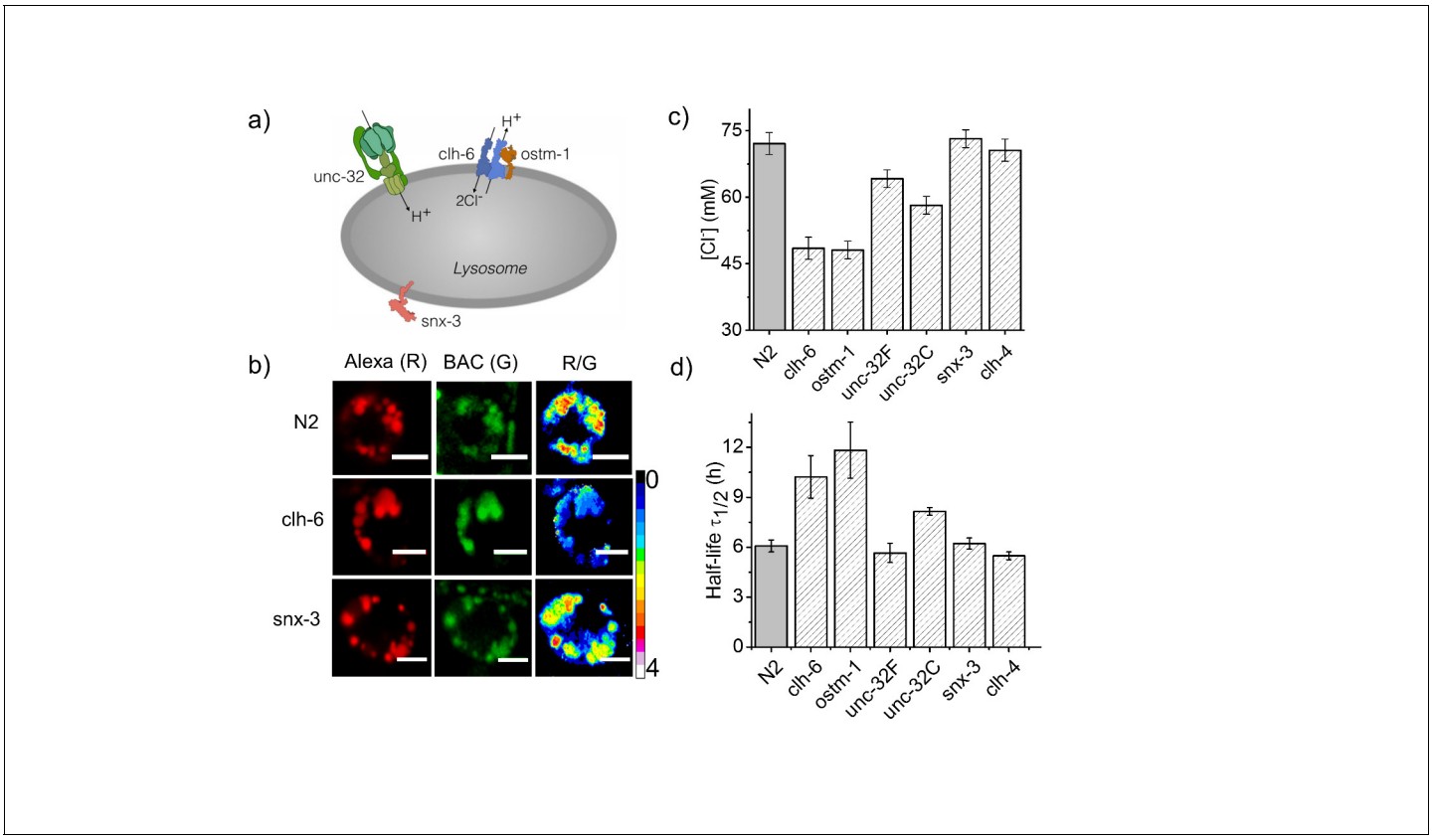

**Figure 2.** Dysregulation in lysosomal [Cl⁻] correlates with reduced lysosomal degradation. (a) Schematic depicting protein players involved in autosomal recessive osteopetrosis. (b) Representative images of *Clensor* in lysosomes of coelomocytes, in the indicated genetic backgrounds acquired in the Alexa 647 (R) and BAC (G) channels and their corresponding pseudocolored R/G images. Scale bar, 5 μm. (c) Lysosomal Cl⁻ concentrations ([Cl⁻]) measured using *Clensor* in indicated genetic background (n = 10 worms, ≥100 lysosomes). (d) Degradative capacity of lysosomes of coelomocytes in nematodes with the indicated genetic backgrounds as given by the observed half-life of *Clensor*. Error bars indicate s.e.m.

The following figure supplements are available for figure 2:

**Figure supplement 1.** (a) Representative images of coelomocyte lysosomes labeled with *Clensor* one hour post injection, in the indicated genetic backgrounds acquired in the Alexa 647 (R) and BAC (G) channels and the corresponding pseudocolored R/G images.

**Figure supplement 2.** (a) Plots showing mean whole cell intensity of *I4*$_{A647}$ per coelomocyte, as a function of time, post-injection in indicated genetic backgrounds.

capacity, inversely correlated with their lysosomal chloride values (*Figure 2d* and *Figure 2—figure supplement 2*).

In this context, data from *snx-3* and *unc-32f* mutants support that high lysosomal chloride is critical to the degradation function of the lysosome. In humans, SNX10 is thought to be responsible for the vesicular sorting of V-ATPase from the Golgi or for its targeting to the ruffled border (*Aker et al., 2012*). Non-functional SNX10 can thus be considered a 'secondary V-ATPase deficiency', phenocopying a V-ATPase deficiency and showing osteoclasts without ruffled borders due to defective lysosomal transport (*Aker et al., 2012*). Importantly, lysosomal pH in *snx-3* knockdowns was compromised by 0.3 pH units, while that in *unc-32* knockdowns was compromised by 0.2 pH units (*Figure 2—figure supplement 1*) (*Chen et al., 2012*). Yet both these genetic backgrounds showed completely normal lysosomal degradation capacity, that is consistent with their normal lumenal chloride levels, rather than their defective pH levels. This further supports that high lysosomal chloride is a sensitive correlate of the degradative function of the lysosome.

## Lysosomal chloride is highly depleted in lysosomal storage disorders

Since lysosomal chloride dysregulation correlated with a loss of degradative ability of the lysosome, we wondered whether the converse was true, i.e., whether lysosomes known to be defective in degradation as seen in lysosomal storage disorders, showed depleted chloride levels. Given that in higher organisms such as mice and humans, high acidity has also been shown to be essential for proper lysosome function (*Mindell, 2012*), we measured both lysosomal pH and lysosomal chloride in *C. elegans* mutants and RNAi knockdowns for a range of genes that are known to cause lysosomal storage disorders. These included a selection of diseases due to dysfunctional enzymes that metabolize sugar derivatives, such as mannose and glycosaminoglycans, as well as lipids such as sphingomyelin and glucosylceramide. Lysosomal pH and chloride measurements were made with $I4^{cLY}$ and

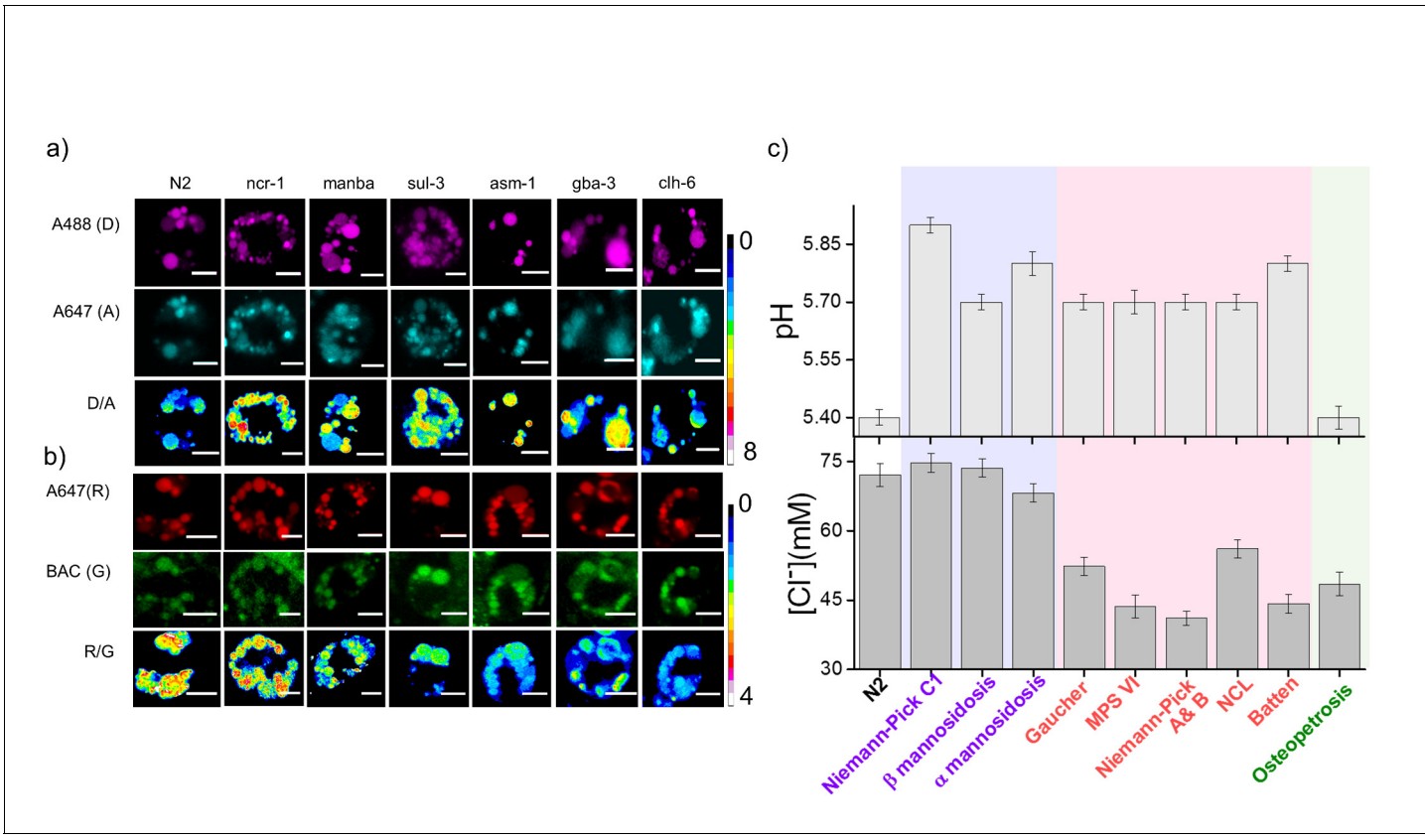

**Figure 3.** Lysosomal chloride dysregulation is observed in nematode models in several pH-related lysosomal storage disorders. (**a**) Representative pH maps of lysosomes in coelomocytes labelled with a DNA-based pH reporter, $I4^{cLY}_{A488/A647}$, in the indicated genetic backgrounds. Images were acquired in the donor (D, magenta) and acceptor (A, cyan) channels and the corresponding pseudocolored D/A images. Scale bar, 5 μm (**b**) Representative [Cl⁻] maps of lysosomes acquired in these genetic backgrounds using *Clensor*. Images are acquired in the Alexa 647 (R) and BAC (G) channels and the corresponding pseudocolored R/G images are shown. Scale bar, 5 μm. (**c**) Quantification of lysosomal pH and lysosomal Cl⁻ in *C. elegans* mutants or RNAi knockdowns of genes responsible for the indicated lysosomal storage diseases in humans. Mutants are grouped according to dysregulation only in lysosomal pH (purple box); only in lysosomal chloride (green box) and both lysosomal pH and chloride (pink box) for n = 10 worms (≥100 lysosomes) Error bars indicate s.e.m.

The following figure supplements are available for figure 3:

**Figure supplement 1.** (**a**) Representative images of LMP-1::GFP marked coelomocytes in the background of indicated RNAi.

**Figure supplement 2.** (**a**) Worms expressing LMP-1::GFP in coelomocytes were injected with $I4^{cLY}_{A647}$ or *Clensor*$_{A647}$ (red) and show maximal colocalization with LMP-1::GFP vesicles (green) at 60 min.

**Figure supplement 3.** (**a**) Histograms comparing the spread of D/A in coelomocytes in different RNAi background.

*Clensor* respectively, in each genetic background at 60 min post injection (*Figure 3a and b*). We found that in *C. elegans* mutants for Gaucher's disease, Batten disease, different forms of NCL, MPS VI and Niemann Pick A/B disease, lysosomal chloride levels were severely compromised (*Figure 3a and b*). Dysfunctional lysosomes showed three types of ion profiles, those where either lysosomal acidity or chloride levels were reduced, and those where both lysosomal acidity and chloride were reduced. The magnitude of proton dysregulation in these defective lysosomes ranged between 1.9–2.8 μM. However, the magnitude of lysosomal chloride showed a stark drop, decreasing by 19–34 mM in most mutants. Importantly, in mammalian cell culture models for many of these diseases example for Gaucher's disease, NCL, MPS VI, etc., only pH dysregulation has been reported (*Bach et al., 1999*; *Holopainen et al., 2001*; *Sillence, 2013*). Yet we find that in *C. elegans* models of these diseases that chloride levels are highly compromised. Chloride decreases by nearly three orders of magnitude more than proton decrease, and the percentage changes of both ions are similar.

To check whether such chloride decrease is observed also in higher organisms, we made pH and chloride measurements in mammalian cell culture models of two relatively common lysosomal storage disorders. Macrophages are a convenient cell culture system to study lysosomal storage disorders as they can be isolated from blood samples and have a lifetime of 3 weeks in culture (*Vincent et al., 1992*). We re-created two widely used murine and human cell culture models of Gaucher's disease by inhibiting β-glucosidase with its well-known inhibitor conduritol β epoxide (CBE) in murine and human macrophages namely, J774A.1 and THP-1 cells respectively (*Hein et al., 2013*, *2007*; *Schueler et al., 2004*). We also recreated common mammalian cell culture models of Niemann-Pick A/B disease by inhibiting acid sphinogomyelinase (SMPD1) in J774A.1 and THP-1 cells with a widely used inhibitor amitriptyline hydrochloride (AH) (*Aldo et al., 2013*; *Jones et al., 2008*). First we confirmed that *Clensor* and our DNA-based pH reporter localized exclusively in lysosomes. In both cell lines, DNA nanodevices (500 nM) were uptaken from the extracellular milieu by the scavenger receptors, followed the endolysosomal pathway and showed quantitative colocalization with lysosomes that were pre-labelled with TMR-Dextran (*Figure 4—figure supplement 3a and b*). In-cell calibration curves of both pH (*Figure 4—figure supplement 1*) and chloride reporters (*Figure 4a*) were well matched with their *in vitro* calibration profiles, indicating that both sensor integrity and performance were quantitatively preserved at the time of making lysosomal pH and chloride measurements in these cells. Both human and murine lysosomes in normal macrophages showed chloride concentrations close to ~118 mM, revealing that lysosomes have the highest chloride levels compared to any other endocytic organelle (*Saha et al., 2015*; *Sonawane et al., 2002*). This is nearly 10–15% higher than even extracellular chloride concentrations, which reaches only up to 105–110 mM (*Arosio and Ratto, 2014*).

Treating J774A.1 cells and THP-1 cells with a global chloride ion channel blocker, such as NPPB (5-Nitro-2-(3-phenylpropylamino) benzoic acid), lowered lysosomal chloride concentrations to 104 and 106 mM respectively, indicating that *Clensor* was capable of measuring pharmacologically induced lysosomal chloride changes, if any, in these cells. In Gaucher's cell culture models, murine and human cells showed a substantial decrease in lysosomal chloride to ~101 mM and ~92 mM respectively. This is a drop of 15–25 mM (13–21% change) chloride, as compared to a drop of ~10 μM in lysosomal proton concentrations. In Niemann-Pick A/B cell culture models, murine and human macrophages showed an even more dramatic decrease in lysosomal chloride to ~77 mM and ~86 mM respectively. This is also a substantial decrease of 30–40 mM (25–34% change) chloride, as compared to a drop of ~9 μM in lysosomal proton concentrations. On average in these four cell culture models, we find that the magnitude of chloride concentration decrease is at least 3 orders of magnitude greater than proton decrease, indicating that lysosome dysfunction is easily and sensitively reflected in its lumenal chloride concentrations. A Niemann Pick C cell culture model using the inhibitor U18666A recapitulated our findings in nematode models, where only lysosomal pH, but not Cl⁻, was altered (*Figure 4—figure supplement 5*)

## High chloride regulates lysosome function in multiple ways

The ClC family protein CLC-7 is expressed mainly in the late endosomes and lysosomes (*Graves et al., 2008*; *Jentsch, 2007*). The loss of either ClC-7 or its β-subunit Ostm1 does not affect lysosomal pH in any way, yet leads to osteopetrosis, resulting in increased bone mass, and severe degeneration of the brain and retina (*Lange et al., 2006*). Along with our studies in nematodes, this

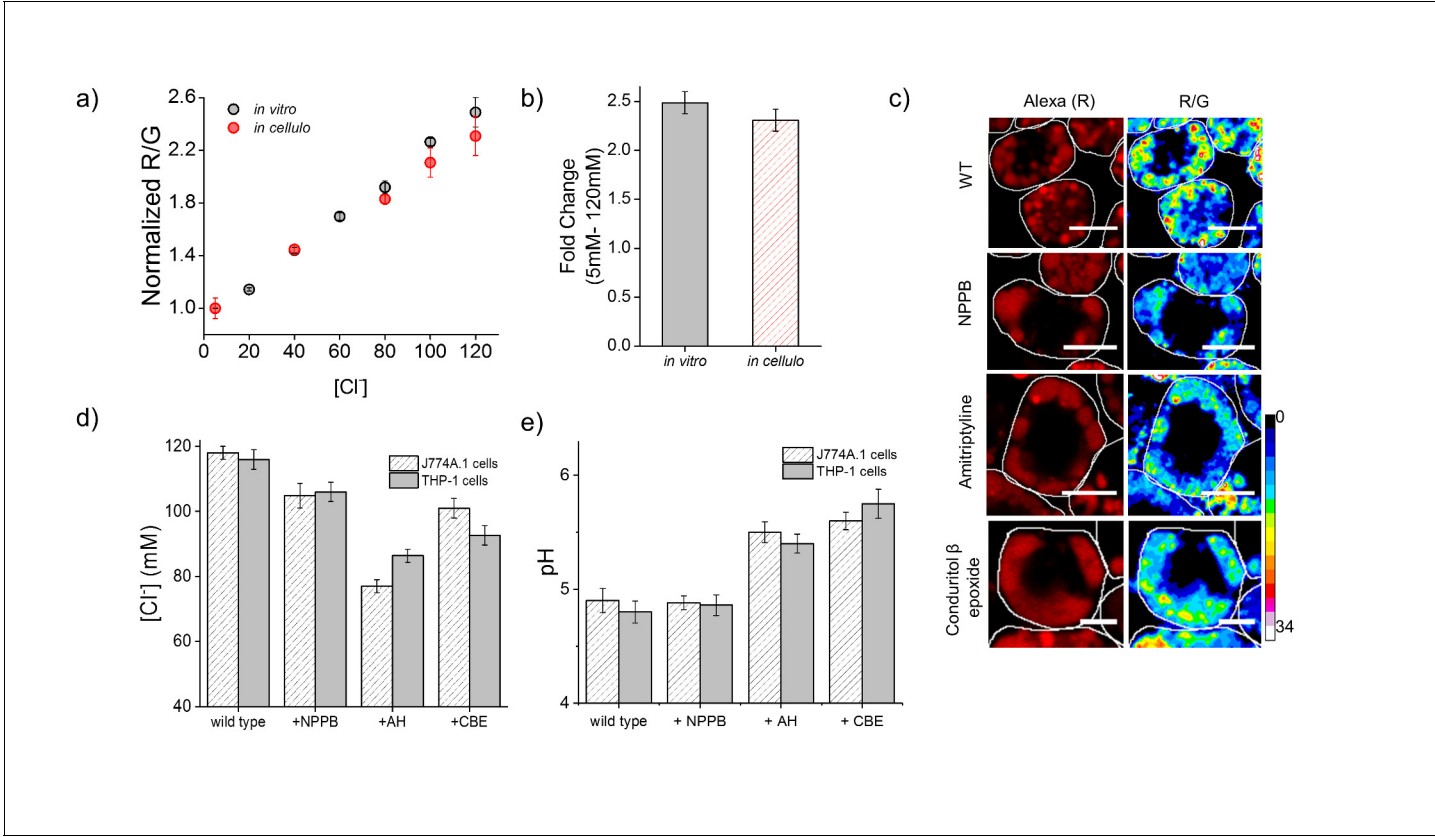

**Figure 4.** Lysosomal chloride is substantially depleted in mammalian cell culture models of lysosomal storage diseases. (a) Calibration profile of *Clensor* in cells (red) and *in vitro* (grey) showing normalized Alexa 647 (R) and BAC (G) intensity (R/G) ratios versus [Cl⁻]. Error bars indicate s.e.m. (n = 20 cells, ≥100 endosomes) (b) Fold change in R/G ratios of *Clensor in vitro* (grey) and in cells (red) from 5 mM to 120 mM [Cl⁻] (c) Representative [Cl⁻] maps of *Clensor* in lysosomes of J774A.1 cells treated with the indicated lysosomal enzyme inhibitor. Images of the Alexa 647 (R) channel and pseudocolored R/G images are shown. Scalebar: 10 μm. (d) Bar graphs of lysosomal Cl⁻ values obtained in THP-1 and J774A.1 cells treated with the indicated inhibitors. NPPB (50 μM), Amitryptiline, AH (10 μM), Conduritol *β*-epoxide, CBE (400 μM) were used to model Niemann Pick A/B and Gaucher's diseases in both cell types. Error bars indicate s.e.m. (n = 10 cells, ≥60 endosomes). (e) Bar graphs of lysosomal pH values obtained in THP-1 and J774A.1 cells treated with the indicated inhibitors. NPPB (50 μM), Amitryptiline, AH (10 μM), Conduritol *β*-epoxide, CBE (400 μM) were used to model Niemann Pick A/B and Gaucher's diseases respectively in both cell types. Error bars indicate s.e.m. (n = 10 cells, ≥50 endosomes).

The following figure supplements are available for figure 4:

**Figure supplement 1.** (a) Structure of Oregon Green (OG) and schematic of I$^{mLy}$ (b) Fluorescence emission spectra of I$^{mLy}$ at the indicated pH obtained using $\lambda_{Ex}$OG = 494 nm (green) and $\lambda_{Ex}$ Atto 647N = 650 nm (red).

**Figure supplement 2.** Plots showing mean whole cell intensity (wci, black line) of *Clensor*$_{A647}$ in cells imaged as a function of indicated times in (a) J774A.1 cells and (b) THP-1 cells.

**Figure supplement 3.** (a) Representative images showing colocalization of I$^{mLy}$$_{AT647}$ with TMR-Dex in J774A.1 and THP-1 macrophages (b) Macrophage labeling efficiency with *Clensor*$_{A647}$ (A647) in the absence or presence of 50 equivalents excess of maleylated bovine serum albumin (mBSA) in comparison to TMR Dextran.

**Figure supplement 4.** Co-localization of *Clensor* (red) with lysosomes labelled with TMR–dextran (green) in J774A.1 cells treated with the indicated lysosomal enzyme inhibitors.

**Figure supplement 5.** Representative pH and [Cl⁻] maps of I$^{mLy}$ and *Clensor* in lysosomes of J774A.1 cells in the absence and presence of 10 μM U18666A that gives a cell culture model of NP-C.

reveals a role for high chloride in lysosome function that is beyond that of a mere counter-ion in the lysosome. We therefore probed whether it could indirectly affect lysosomal function by affecting lysosomal Ca$^{2+}$ (*Luzio et al., 2007*; *Rodr?guez et al., 1997*; *Shen et al., 2012*). We also considered the possibility that lysosomal chloride could exert a direct effect, where its reduction could impede the function of lysosomal enzymes thus affecting its degradative capacity (*Baccino et al., 1975*; *Cigic and Pain, 1999*; *Maurus et al., 2005*; *Wartosch and Stauber, 2010*) (*Figure 5a*).

Lysosomes are also intracellular Ca$^{2+}$ stores with free Ca$^{2+}$ ranging between ~400–600 µM (*Christensen et al., 2002*; *Lloyd-Evans et al., 2008*). The principal Ca$^{2+}$ channel responsible for lysosomal Ca$^{2+}$ release is Mucolipin TRP channel 1 (TRPML1). We therefore sought to estimate lysosomal Ca$^{2+}$ by measuring Ca$^{2+}$ that is released from the lysosome using two different triggers under conditions of normal and reduced lysosomal Cl$^-$. Glycyl-L-phenylalanine 2-naphthylamide (GPN) is a substrate for Cathepsin C, which when added to cells, gets cleaved in the lysosome to release

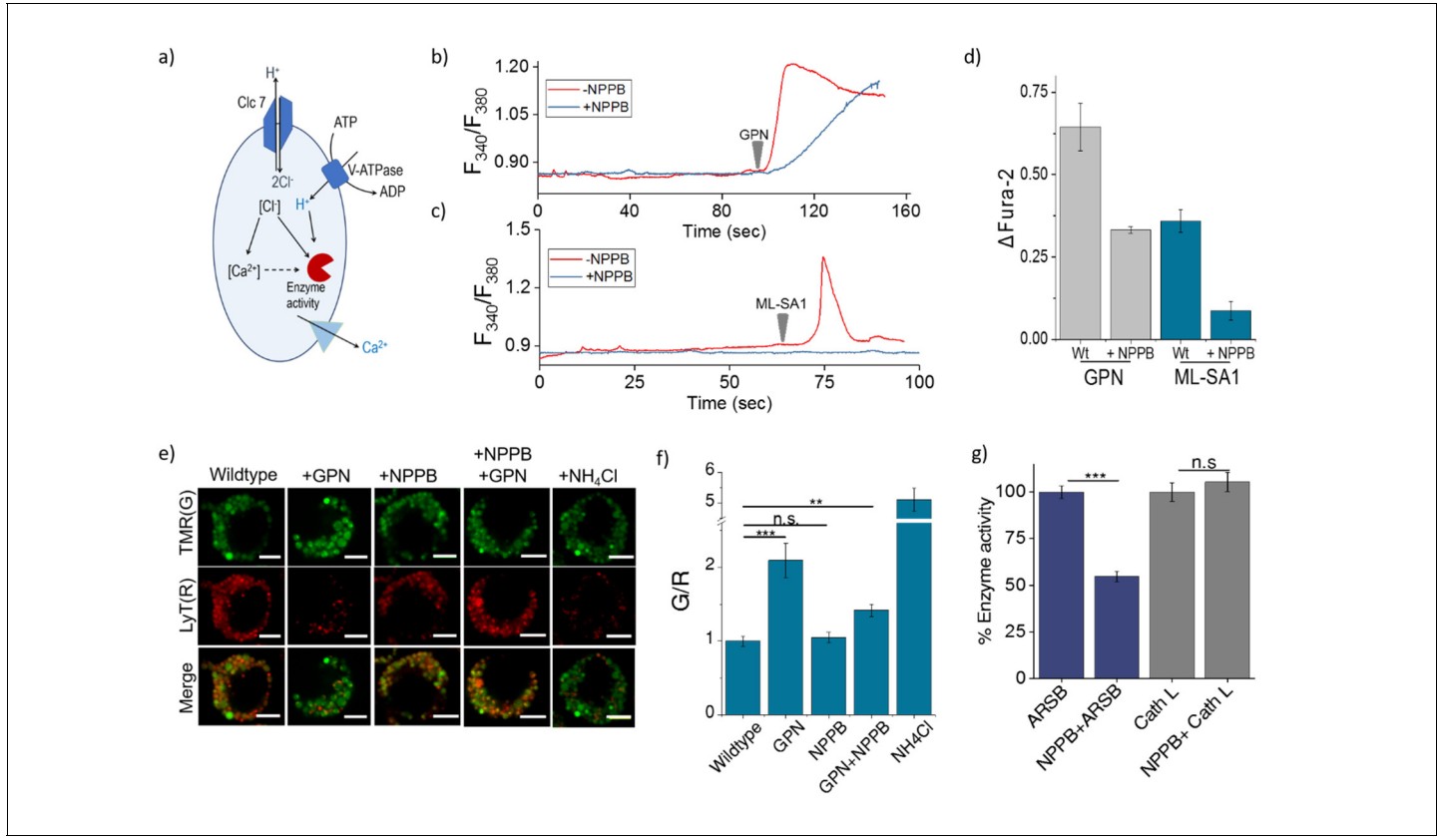

**Figure 5.** (a) Schematic of potential roles for lysosomal chloride. Cl$^-$ ions can regulate lysosomal Ca$^{2+}$ and/or affect lysosomal enzyme function. (b) Representative traces of Glycyl-L-phenylalanine 2-naphthylamide (GPN) (400 µM) triggered lysosomal Ca$^{2+}$ release in J774A.1 cells ratiometrically imaged using Fura-2 ($F_{340}/F_{380}$) in the presence and absence of 50 µM NPPB. (c) Representative traces of ML-SA1 (20 µM) triggered lysosomal Ca$^{2+}$ release in *J774A.1* cells ratiometrically imaged using Fura-2 ($F_{340}/F_{380}$) in the presence and absence of 50 µM NPPB. (d) Quantification of lysosomal Ca$^{2+}$ release from b) and c) given by ($F_t$-$F_0/F_0$) (ΔFura-2) for n = 15 cells. (e) Representative images of lysosomes of J774A.1 cells labelled with TMR dextran (TMR; **G**) and LysoTracker Red (LyT; **R**) in the presence or absence of 50 µM NPPB, 200 µM GPN or 1 mM NH$_4$Cl. Scale bar, 5 µm (f) Quantification of LysoTracker Red release from (e) (n = 50 cells). (**g**) Quantification of activity of the enzymes arylsulfatase B (ARSB) and Cathepsin L (Cath L) in J774A.1 cells in the absence and presence of 50 µM NPPB (n = 70 cells). Error bars indicate s.e.m. P values are as follows; ** = p<0.001, *** = p<0.0001, n.s = non significant.

The following figure supplements are available for figure 5:

**Figure supplement 1.** Representative fluorescence images of cleaved substrates of ARSB and cathepsin L (E), DAPI (D) merge of E and D channels and respective pseudocolour E/D maps of *J774A.1* cells with and without 50 µM NPPB.

**Figure supplement 2.** (a) Lysosomal pH and (b) chloride levels measured by I$^{mLy}$ and *Clensor* in J774A.1 cells with increasing concentrations of NPPB.

naphthylamine that is known to compromise the integrity of the lysosomal membrane, leading to a leakage of ions such as $Ca^{2+}$ into the cytosol (*Berg et al., 1994*; *Jadot et al., 1984*; *Morgan et al., 2011*). This has been used to induce lysosomal $Ca^{2+}$ release.

The cytosol of J774A.1 cells are labeled with ~3 µM Fura2-AM to ratiometrically image cytosolic $Ca^{2+}$ elevation upon its release, if at all, from the lysosome. After addition of 400 µM GPN, cells were continuously imaged ratiometrically over 15–20 mins. Shortly after GPN addition, a burst of $Ca^{2+}$ was observed in the cytosol, corresponding to released lysosomal $Ca^{2+}$ (*Figure 5b*). When the same procedure was performed on cells that had been incubated with 50 µM NPPB that reduces lysosomal Cl-, the amount of lysosomal $Ca^{2+}$ released was significantly reduced (*Figure 5b–d*) We then performed a second, more targeted way to release lysosomal $Ca^{2+}$ into the cytosol, by using 20 µM ML-SA1 which specifically binds to and opens the TRPML1 channel on lysosomes (*Shen et al., 2012*). We found that when lysosomal Cl- was reduced with NPPB, lysosomal $Ca^{2+}$ release into the cytosol was near negligible (*Figure 5c–d*). Taken together this indicates that high lysosomal Cl- is necessary for effective lysosomal $Ca^{2+}$ release, possibly by affect lysosomal $Ca^{2+}$ accumulation.

We next investigated whether reducing lysosomal chloride directly impacted the activity of any lysosomal enzymes. *In vitro* enzymology of Cathepsin C, a lysosome-resident serine protease has revealed that increasing Cl- increased its enzymatic activity (*Cigic and Pain, 1999*; *McDonald et al., 1966*). Further, the crystal structure of Cathepsin C shows bound chloride ions close to the active site (*Cigic and Pain, 1999*; *Turk et al., 2012*). We therefore used GPN cleavage to probe Cathepsin C activity in the lysosome upon reducing Cl- with NPPB. GPN cleavage by Cathepsin C releases naphthylamine which compromises lysosomal membrane integrity leading to proton leakage from the lysosome into the cytosol. This hypoacidifies the lysosomes resulting in reduced LysoTracker labeling as the labeling efficiency of the latter is directly proportional to compartment acidity.

Lysosomes are pre-labeled with TMR-Dextran, and LysoTracker intensities are normalized to the fluorescence intensity of TMR-Dextran, given as G/R. Hypoacidifying lysosomes by addition of 1 mM $NH_4Cl$ indeed reduced LysoTracker labeling, as expected (*Figure 5e–f*). A similar effect was also obtained upon GPN addition. The presence or absence of NPPB showed no change in LysoTracker labeling in cells (*Figure 5e–f*), indicating that NPPB by itself caused no alteration in lysosomal pH. However, when GPN was added to NPPB treated cells LysoTracker staining was remarkably well preserved (*Figure 5e and f*) indicating preservation of lysosomal membrane integrity because GPN was no longer effectively cleaved by Cathepsin C when lysosomal Cl- was reduced. Unlike other cathepsins, Cathepsin C does not undergo autoactivation but requires processing by Cathepsin L and Cathepsin S to convert it into active Cathepsin C (*Dahl et al., 2001*). We measured the activity of the upstream cathepsins such as Cathepsin L using fluorogenic substrates in the presence and absence of NPPB (*Figure 5g*, *Figure 5—figure supplement 1*). We observed no effect of chloride levels on Cathepsin L activity. This indicates that low Cathepsin C activity is not due to decreased amounts of mature Cathepsin C in the lysosome, but rather, reduced activity of mature Cathepsin C (*Figure 5g*, *Figure 5—figure supplement 1*).

Based on reports suggesting that arylsulfatase B activity was also affected by low chloride (*Wojczyk, 1986*), we similarly investigated a fluorogenic substrate for arylsulfatase and found that NPPB treatment impeded arylsulfatase cleavage in the lysosome. Taken together, these results suggest that high lysosomal chloride is integral to the activity of key lysosomal enzymes and that reducing lysosomal chloride affects their function.

## Conclusions

The lysosome is the most acidic organelle within the cell. This likely confers on it a unique ionic microenvironment, reinforced by its high lumenal chloride, that is critical to its function (*Xu and Ren, 2015*). Using a DNA-based, fluorescent reporter called *Clensor* we have been able to create quantitative, spatial maps of chloride *in vivo* and measured lysosomal chloride. We show that, in *C. elegans*, lysosomes are highly enriched in chloride and that when lysosomal chloride is depleted, the degradative function of the lysosome is compromised. Intrigued by this finding, we explored the converse: whether lysosomes that had lost their degradative function – as seen in lysosomal storage disorders - showed lower lumenal chloride concentrations. In a host of *C. elegans* models for various lysosomal storage disorders, we found that this was indeed the case. In fact, the magnitude of change in chloride concentrations far outstrips the change in proton concentrations by at least three orders of magnitude.

To see whether chloride dysregulation correlated with lysosome dysfunction more broadly, we studied murine and human cell culture models of Gaucher's disease, Niemann-Pick A/B disease and Niemann Pick C. We found that in mammalian cells too, lysosomes are particularly rich in chloride, surpassing even extracellular chloride levels. Importantly, chloride values in all the mammalian cell culture models revealed magnitudes of chloride dysregulation that were similar to that observed in *C. elegans*.

Our findings suggest more widespread and as yet unknown roles for the single most abundant, soluble physiological anion in regulating lysosome function. Decrease in lysosomal chloride impedes the release of calcium from the lysosome implicating an interplay between these two ions in the lysosome. It is also possible that chloride accumulation could facilitate lysosomal calcium enrichment through the coupled action of multiple ion channels. The ability to quantitate lysosomal chloride enables investigations into the broader mechanistic roles of chloride ions in regulating multiple functions performed by the lysosome. As such, given that chloride dysregulation shows a much higher dynamic range than hypoacidification, quantitative chloride imaging can provide a much more sensitive measure of lysosome dysfunction in model organisms as well as in cultured cells derived from blood samples that can be used in disease diagnoses and screening applications.

## Materials and methods

### Reagents

All fluorescently labeled oligonucleotides were HPLC-purified and obtained from IBA-GmBh (Germany) and IDT (Coralville, IA, USA). Unlabeled oligonucleotides were purchased from IDT (Coralville, IA, USA). The peptide nucleic acids (PNA) oligomer, P was synthesized using standard solid phase Fmoc chemistry on Nova Syn TGA resin (Novabiochem, Germany) using analytical grade reagents (Applied Biosystems, USA), purified by reverse phase HPLC (Shimadzu, Japan) as previously reported and stored at −20°C until further use (*Prakash et al., 2016*).

Bovine serum albumin (66 kDalton), nigericin, valinomycin, monensin, chloride ionophore I, **Isopropyl β-D-1-thiogalactopyranoside** (IPTG), amitriptyline hydrochloride, 5-nitro-2-(3-phenylpropylamino) benzoic acid (NPPB) and conduritol $\beta$ epoxide (CBE) were obtained from Sigma (USA). LysoTracker Deep Red, TMR-Dextran (10 kDa) and Oregon Green 488 maleimide was obtained from Molecular Probes, Invitrogen (USA). Lysosomal enzyme kits namely lysosomal sulfatase assay kit was purchased from Marker Gene (USA); Magic Red Cathepsin L assay kit from Immunochemistry Technologies. Gly-Phe $\beta$-naphthylamide was purchased from Santa Cruz Biotechnology (USA). All other reagents were purchased from Sigma-Aldrich (USA) unless otherwise specified. BSA was maleylated according to a previously published protocol (*Haberland and Fogelman, 1985*). Trizol was purchased from Invitrogen (U.S.A.).

### Sample preparation

All oligonucleotides were ethanol precipitated and quantified by their UV absorbance. For I-switch ($I4^{cLY}_{A488/A647}$) sample preparation, 5 μM of I4 and I4′ were mixed in equimolar ratios in 20 mM potassium phosphate buffer, pH 5.5 containing 100 mM KCl. The resulting solution was heated to 90°C for 5 min, cooled to the room temperature at 5°C/15 mins and equilibrated at 4°C overnight. Samples were diluted and used within 7 days of annealing. A sample of *Clensor* was similarly prepared using HPLC purified oligonucleotides and PNA oligomer at a final concentration of 10 μM by mixing D1, D2 and P (see Table S1 for sequence information) in equimolar ratios in 10 mM sodium phosphate buffer, pH 7.2 and annealed as described above. For $I^{mLy}$, Oregon Green maleimide was first conjugated to the thiol labeled oligonucleotide (*Hermanson, 2008*). Briefly, to 10 μM thiol labelled oligonucleotide in HEPES pH 7.4, 500 μM of TCEP (tris-carboxyethylphosphine) was added to reduce the disulfide bonds. After 1 hr at room temperature, 50 μM Oregon Green Maleimide was added and the reaction was kept overnight at room temperature. The reaction mixture was purified using an Amicon cutoff membrane filter (3 kDa, Millipore) to remove unreacted dye (*Figure 4—figure supplement 1*). A sample of $I^{mLy}$ was similarly prepared using HPLC purified oligonucleotides at a final concentration of 5 μM by mixing $I^{mLY}_{OG}$ and $I^{mLY}_{AT647}$ (see Table S1 for sequence information) in equimolar ratios in 10 mM sodium phosphate buffer, pH 7.2 and annealed as described

above. Prior to use, all buffer stock solutions were filtered using 0.22 μm disk filters (Millipore, Germany).

## C. elegans methods and strains

Standard methods were followed for the maintenance of *C. elegans*. Wild type strain used was the *C. elegans* isolate from Bristol, strain N2 (*Brenner, 1974*). Strains used in the study, provided by the Caenorhabditis Genetics Center (CGC), are RRID:WB-STRAIN:RB920 *clh-6(ok791)*, RRID:WB-STRAIN:FF451 *unc-32(f131)*, RRID:WB-STRAIN:CB189 *unc-32(e189)*, RRID:WB-STRAIN:MT7531 *ppk-3(n2835)*, RRID:WB-STRAIN:VC3135 *gba-3(gk3287)*, RRID:WB-STRAIN:VC183 *ppt-1(gk139)*, and RRID:WB-STRAIN:XT7 *cln-3.2(gk41) I; cln-3.3(gk118) cln-3.1(pk479)*. Transgenics used in this study, also provided by the CGC, are RRID:WB-STRAIN:GS1912 *arIs37 [pmyo-3::ssGFP]*, a transgenic strain that expresses ssGFP in the body wall muscles, which is secreted in the pseudocoelom and endocytosed by coelomocytes and RRID:WB-STRAIN:RT258 *pwIs50 [lmp-1::GFP + Cb-unc-119(+)]*, a transgenic strain expressing GFP-tagged lysosomal marker LMP-1. Genes, for which mutants were unavailable, were knocked down using Ahringer library based RNAi methods (*Kamath and Ahringer, 2003*). The RNAi clones used were: *L4440* empty vector control, *ncr-1* (clone F02E8.6, Ahringer Library), *ostm1* (clone F42A8.3, Ahringer Library), *snx-3* (clone W06D4.5, Ahringer Library), *manba* (clone C33G3.4, Ahringer Library), *aman-1* (clone F55D10.1, Ahringer Library), *sul-3* (clone C54D2.4, Ahringer Library), *gba-3* (clone F11E6.1, Ahringer Library) and *asm1* (clone B0252.2, Ahringer Library).

## Coelomocyte labeling experiments

Coelomocyte labeling and competition experiments were carried out with I4$^{cLY}_{A647}$, and *Clensor*$_{A647}$ as described previously by our lab (*Surana et al., 2011*). For microinjections, the samples were diluted to 100 nM using 1X Medium 1 (150 mM NaCl, 5 mM KCl, 1 mM CaCl$_2$, 1 mM MgCl$_2$, 20 mM HEPES, pH 7.2). Injections were performed, in the dorsal side in the pseudocoelom, just opposite to the vulva, of one-day old wild type hermaphrodites using an Olympus IX53 Simple Inverted Microscope (Olympus Corporation of the Americas, Center Valley, PA) equipped with 40X, 0.6 NA objective, and microinjection setup (Narishige, Japan). Injected worms were mounted on 2.0% agarose pad and anesthetized using 40 mM sodium azide in M9 buffer. In all cases labeling was checked after 1 hr incubation at 22°C.

## Colocalization experiments

I4$^{cLY}_{A647}$ or *Clensor*$_{A647}$ sample was diluted to 100 nM using 1X Medium 1 and injected in 10 *arIs37 [pmyo-3::ssGFP]* hermaphrodites as described previously by our lab (*Surana et al., 2011*). Imaging and quantification of the number of coelomocytes labeled, after 1 hr of incubation, was carried out on the Leica TCS SP5 II STED laser scanning confocal microscope (Leica Microsystems, Inc., Buffalo Grove, IL) using an Argon ion laser for 488 nm excitation and He-Ne laser for 633 excitation with a set of dichroics, excitation, and emission filters suitable for each fluorophore. Cross talk and bleed-through were measured and found to be negligible between the GFP/Alexa 488/BAC channel and Alexa 647 channel.

## RNAi experiments

Bacteria from the Ahringer RNAi library expressing dsRNA against the relevant gene was fed to worms, and measurements were carried out in one-day old adults of the F1 progeny (*Kamath and Ahringer, 2003*). RNA knockdown was confirmed by probing mRNA levels of the candidate gene, assayed by RT-PCR. Briefly, total RNA was isolated using the Trizol-chloroform method; 2.5 μg of total RNA was converted to cDNA using oligo-dT primers. 5 μL of the RT reaction was used to set up a PCR using gene-specific primers. Actin mRNA was used as a control. PCR products were separated on a 1.5% agarose-TAE gel. Genes in this study that were knocked down by RNAi correspond to clh-6, ncr-1 and ostm-1 that showed expected 1.1 kb (clh-6); 1.1 kb (ncr-1); 0.9 kb (ostm-1) etc (*Figure 1—figure supplement 1*).

## *In vitro* fluorescence measurements

Fluorescence spectra were measured on a FluoroMax-4 Scanning Spectrofluorometer (Horiba Scientific, Edison, NJ, USA) using previously established protocols (*Modi et al., 2009*; *Saha et al., 2015*). Briefly, I4$^{cLY}_{A488/A647}$ was diluted to 50 nM in 1X pH clamping buffer of desired pH for all *in vitro* fluorescence experiments. All samples were vortexed and equilibrated for 30 min at room temperature. The samples were excited at 488 nm and emission collected between 505–750 nm. A calibration curve was obtained by plotting the ratio of donor emission intensity (D) at 520 nm and acceptor intensity (A) at 669 nm (for A488/A647) as a function of pH. Mean of D/A from three independent experiments and their s.e.m were plotted for each pH value. For *in vitro* calibration of I$^{mLy}$, 50 nM of the sensor is diluted into 1X pH clamping buffer of desired pH. Oregon Green and Atto 647N are excited at 490 nm and 645 nm respectively. Emission spectra for Oregon Green and Atto 647N were collected between 500–550 nm and 650–700 nm respectively. A calibration curve was obtained by plotting the ratio of Oregon Green (G) at 520 nm and Atto 647N (R) at 665 nm (for G/R) as a function of pH. Mean of G/R from three independent experiments and their s.e.m were plotted for each pH value.

For chloride measurements, 10 µM stock of *Clensor* was diluted to a final concentration of 200 nM using 10 mM sodium phosphate buffer, pH 7.2 and incubated for 30 min at room temperature prior to experiments. BAC and Alexa 647 were excited at 435 nm for BAC and 650 nm for Alexa 647 respectively. Emission spectra of BAC and Alexa 647 were collected between 495–550 nm and 650–700 nm respectively. In order to study the chloride sensitivity of *Clensor*, final chloride concentrations ranging between 5 mM to 80 mM were achieved by addition of microliter aliquots of 1 M stock of NaCl to 400 µL of sample. Emission intensity of BAC at 505 nm (G) was normalized to emission intensity of Alexa 647 at 670 nm (R). Fold change in R/G ratio was calculated from the ratio of R/G values at two specific values of [Cl], either 5 mM and 80 mM or 5 mM and 120 mM as mentioned in the text.

## *In vivo* measurements of pH and chloride pH

Clamping and real time measurement experiments were carried out with I4$^{cLY}_{A488/A647}$ as described by our lab previously (*Modi et al., 2009*; *Surana et al., 2011*). For microinjections, the I-switch sample was diluted to 500 nM using 1X Medium 1. Briefly, worms were incubated at 22°C for 1 hr post microinjection and then immersed in clamping buffers (120 mM KCl, 5 mM NaCl, 1 mM MgCl$_2$, 1 mM CaCl$_2$, 20 mM HEPES) of varying pH, containing 100 µM nigericin and 100 µM monensin. In order to facilitate entry of the buffer into the body, the cuticle was perforated at three regions of the body using a microinjection needle. After 75 mins incubation in the clamping buffer, coelomocytes were imaged using wide field microscopy. Three independent measurements, each with 10 worms, were made for each pH value.

Chloride clamping and real time measurements were carried out using *Clensor*. Worms were injected with 2 µM of *Clensor* and incubated at 22°C for 2 hr. To obtain the chloride calibration profile, the worms were then immersed in the appropriate chloride clamping buffer containing a specific concentration of chloride, 100 µM nigericin, 100 µM valinomycin, 100 µM monensin and 10 µM chloride ionophore I for 45 mins at room temperature. Chloride calibration buffers containing different chloride concentrations were prepared by mixing the 1X chloride positive buffer (120 mM KCl, 20 mM NaCl, 1 mM CaCl$_2$, 1 mM MgCl$_2$, 20 mM HEPES, pH, 7.2) and 1X chloride negative buffer (120 mM KNO$_3$, 20 mM NaNO$_3$, 1 mM Ca(NO$_3$)$_2$, 1 mM Mg(NO$_3$)$_2$, 20 mM HEPES, pH 7.2) in different ratios.

For real-time lysosomal pH or chloride measurements, 10 hermaphrodites were injected with I4$^{cLY}_{A488/A647}$ or *Clensor* respectively and incubated at 22°C for 1 hr. Worms were then anaesthetized and imaged on a wide field inverted microscope for pH measurements and confocal microscope for chloride measurements.

## Cell culture methods and maintenance

Mouse alveolar macrophage J774A.1 cells were a kind gift from Prof Deborah Nelson, Department of Pharmacological and Physiological Sciences, the University of Chicago, cultured in Dulbecco's Modified Eagle's Medium/F-12 (1:1) (DMEM-F12) (Invitrogen Corporation,USA) containing 10% heat inactivated Fetal Bovine Serum (FBS) (Invitrogen Corporation, USA). THP-1 monocyte cell line was

obtained from late Professor Janet Rowley's Lab at the University of Chicago. Cells were cultured in RPMI 1640 containing 10% heat-inactivated FBS, 10 mM HEPES, 2 mM glutamine, 100 U/ml penicillin, and 100 µg/ml streptomycin, and maintained at 37°C under 5% CO2. All reagents and medium were purchased from (Invitrogen Corporation,USA). THP-1 monocytic cells were differentiated into macrophages in 60 mm dishes containing 3 ml of the RPMI 1640 medium containing 10 nM PMA over 48 hr. These cells are not on the list of commonly misidentified cell lines maintained by the International Cell Line Authentication Committee. The sources of each cell line used in this study are as mentioned above and were used directly by us without additional authentication beyond that provided by the sources. All cells were regularly checked for mycoplasma contamination and were found to be negative for contamination as assayed by DAPI staining.

### *In cellulo* measurements pH and chloride

Chloride clamping and measurements were carried out using *Clensor* using a previously published protocol from our lab (*Saha et al., 2015*). J774A.1 and THP-1 cells were pulsed and chased with 2 µM of *Clensor*. Cells are then fixed with 200 µL 2.5% PFA for 2 min at room temperature, washed three times and retained in 1X PBS. To obtain the intracellular chloride calibration profile, perfusate and endosomal chloride concentrations were equalized by incubating the previously fixed cells in the appropriate chloride clamping buffer containing a specific concentration of chloride, 10 µM nigericin, 10 µM valinomycin, and 10 µM tributyltin chloride (TBT-Cl) for 1 hr at room temperature.

Chloride calibration buffers containing different chloride concentrations were prepared by mixing the 1X chloride positive buffer (120 mM KCl, 20 mM NaCl, 1 mM $CaCl_2$, 1 mM $MgCl_2$, 20 mM HEPES, pH, 7.2) and 1X - chloride negative buffer (120 mM $KNO_3$, 20 mM $NaNO_3$, 1 mM $Ca(NO_3)_2$, 1 mM $Mg(NO_3)_2$, 20 mM HEPES, pH 7.2) in different ratios.

For real-time chloride measurements, cells are pulsed with 2 µM of *Clensor* followed by a 60 min chase. Cells are then washed with 1X PBS and imaged. To see whether *Clensor* can detect changes in Cl accumulation under perturbed conditions, we treated cells with 50 µM NPPB, which is a well-known non-specific Cl channel blocker. Cells were labeled with 2 µM *Clensor* for 30 mins and chased for 30 mins at 37°C. The cells were then chased for 30 mins in media containing 50 µM NPPB and then imaged.

To estimate the chloride accumulation in the lysosomes of Gaucher's Disease in two cell models that is murine J774A.1 and human THP-1 cells, glucosylceramide storage was induced catalytically inactivating the enzyme acid $\beta$-glucosidase, using its well-known inhibitor conduritol $\beta$ epoxide (CBE) (*Grabowski et al., 1986*; *Schueler et al., 2004*). These are both well-documented murine and human cell culture models of Gaucher's disease. Macrophage cells were cultured with 400 µM CBE for 48 hr. Cells were then pulsed and chased with 2 µM *Clensor* as previously described.

To estimate chloride accumulation in the lysosomes of Niemann Pick A/B disease, the same murine and human cell lines were used, and the activity of acid sphingomyelinase (ASM) in these macrophage cell lines was inhibited using the well-known inhibitor, amitriptyline hydrochloride (*Beckmann et al., 2014*; *Kornhuber et al., 2010*). Cells were labeled with 2 µM *Clensor* for 30 mins and chased for 30 mins at 37°C. The cells were then chased for 30 mins in media containing 10 µM amitriptyline hydrochloride and then imaged.

*In cellulo* pH clamping and measurement experiments were carried out with I$^{mLy}$ with modifications to protocols described by our lab previously (*Modi et al., 2013*, *2009*). J774A.1 and THP-1 cells were pulsed and chased with 500 nM of I$^{mLy}$. Cells are then fixed with 200 µL 2.5% PFA for 20 mins at room temperature, washed three times and retained in 1X PBS. To obtain the intracellular pH calibration profile, perfusate and endosomal pH were equalized by incubating the previously fixed cells in the appropriate pH clamping buffer clamping buffers (120 mM $KNO_3$, 5 mM $NaNO_3$, 1 mM $Mg(NO_3)_2$, 1 mM $Ca(NO_3)_2$, 20 mM HEPES, MES and NaOAc) of varying pH, containing 25 µM nigericin and 25 µM monensin for 30 mins at room temperature.

For real-time pH measurements, cells are pulsed with 500 nM of I$^{mLy}$ followed by a 60 mins chase. Cells are then washed with 1X PBS and imaged. pH measurements in the lysosomes of Gaucher's Disease and of Niemann Pick A/B disease, in the two cell models that is murine J774A.1 and human THP-1 cells, were carried out similar to the protocol above using 500 nM of I$^{mLy}$.

## Microscopy

Wide field microscopy was carried out on IX83 research inverted microscope (Olympus Corporation of the Americas, Center Valley, PA, USA) using a 60X, 1.42 NA, phase contrast oil immersion objective (PLAPON, Olympus Corporation of the Americas, Center Valley, PA, USA) and Evolve Delta 512 EMCCD camera (Photometrics, USA). Filter wheel, shutter and CCD camera were controlled using Metamorph Premier Ver 7.8.12.0 (Molecular Devices, LLC, USA), suitable for the fluorophores used. Images on the same day were acquired under the same acquisition settings. All the images were background subtracted taking mean intensity over an adjacent cell free area. Mean intensity in each endosome was measured in donor (D) and acceptor (A) channels. Alexa 488 channel images (D) were obtained using 480/20 band pass excitation filter, 520/40 band pass emission filter and a 89016- ET - FITC/Cy3/Cy5 dichroic filter. Alexa 647 channel images (A) were obtained using 640/30 band pass excitation filter, 705/72 band pass emission filter and 89016- ET - FITC/Cy3/Cy5 dichroic filter. For FRET channel images were obtained using the 480/20 band pass excitation filter, 705/72 band pass emission filter and 89016- ET - FITC/Cy3/Cy5 dichroic filter. Mean intensity in each endosome was measured in donor and acceptor channels. A ratio of donor to acceptor intensities (D/A) was obtained from these readings. Pseudocolor images were generated by calculating the D/A ratio per pixel. Confocal images were captured with a Leica TCS SP5 II STED laser scanning confocal microscope (Leica Microsystems, Inc., Buffalo Grove, IL, USA) equipped with 63X, 1.4 NA, oil immersion objective. Alexa 488 was excited using an Argon ion laser for 488 nm excitation, Alexa 647 using He-Ne laser for 633 excitation and BAC using Argon ion laser for 458 nm excitation with a set of dichroics, excitation, and emission filters suitable for each fluorophore.

Ratiometric calcium imaging of Fura-2 was carried out on an Olympus IX81 microscope equipped with a 40x objective, NA = 1.2. Excitation of Fura-2 is performed using 340/26 and 380/10 nm excitation filters, equipped with a 455 nm dichroic mirror and a 535/40 nm emission filter. Exposure time was kept at 100 ms for all the imaging experiments to minimize phototoxicity.

## Image analysis

Images were analyzed with ImageJ ver 1.49 (NIH, USA). For pH measurements Alexa 488 and Alexa 647 images were overlapped using ImageJ and endosomes showing colocalization were selected for further analysis. Intensity in each endosome was measured in donor (D) and FRET (A) channels and recorded in an OriginPro Sr2 b9.2.272 (OriginLab Corporation, Northampton, MA, USA) file from which D/A ratio of each endosome was obtained. The mean D/A of each distribution were converted to pH according to the intracellular calibration curve. Data was represented as mean pH value ± standard error of the mean. Data for pH clamping experiments was analysed similarly.

For chloride measurements, regions of cells containing lysosomes in each Alexa 647 (R) image were identified and marked in the ROI plugin in ImageJ. The same regions were identified in the BAC (G) image recalling the ROIs and appropriate correction factor for chromatic aberration if necessary. After background subtraction, intensity for each endosome was measured and recorded in an Origin file. A ratio of R to G intensities (R/G) was obtained from these values by dividing the intensity of a given endosome in the R image with the corresponding intensity in the G image. For a given experiment, mean [Cl] of an organelle population was determined by converting the mean R/G value of the distribution to [Cl] values according to the intracellular calibration profile. Data was presented as mean of this mean [Cl] value ± standard error of the mean. Data for chloride clamping experiments was analyzed similarly.

Colocalization of GFP and Alexa 647 was determined by counting the numbers of Alexa 647 positive puncta that colocalize with GFP and representing it as a Pearson's correlation coefficient.

## Lysosomal labelling in coelomocytes

Temporal mapping of I-switch and *Clensor* was done in 10 worms of *pwIs50 [lmp-1::GFP + Cb-unc-119(+)]* as previously described by our lab (*Surana et al., 2011*). Briefly, worms were injected with 500 nM of I4$^{cLY}_{A647}$ or *Clensor*$_{A647}$, incubated at 22℃ for 1 hr, and then imaged using Leica TCS SP5 II STED laser scanning confocal microscope (Leica Microsystems, Inc., Buffalo Grove, IL, USA). Colocalization of GFP and I4$^{cLY}_{A647}$ or *Clensor*$_{A647}$ was determined by counting the numbers of Alexa647 positive puncta that colocalize with GFP positive puncta and expressing them as a percentage of the total number of Alexa 647 positive puncta. In order to confirm lysosomal labeling in a given genetic

background, the same procedure was performed on the relevant mutant or RNAi knockdown in *pwIs50 [lmp-1::GFP + Cb-unc-119(+)]*.

## Statistics and general methods

All pH and chloride clamping experiments (*Figure 1b*, *Figure 1—figure supplement 2*, *Figure 4—figure supplement 2*) were performed in triplicates and the standard error of mean (s.e. m) values are plotted with the number of cells considered being mentioned in each legend. Experiment with murine macrophage, J774A.1 and THP-1 cells (*Figure 4*) has been performed in triplicates. Ratio of standard error of the mean is calculated for n = 20 cells and n = 10 cells and is plotted in *Figure 4d and e* respectively. All pH and chloride measurements in *C.elegans* of indicated genetic backgrounds (*Figures 2c* and *3c* and *Figure 2—figure supplement 1c* ) were carried out in n = 10 worms and the standard error of mean (s.e.m) values are plotted with the number of cells considered being mentioned in each legend.

## DNA stability assay

Coelomocyte labeling for stability assay were carried out with $I4^{cLY}_{A647}$, and $Clensor_{A647}$. For microinjections, the samples were diluted to 500 nM using 1X Medium 1 (150 mM NaCl, 5 mM KCl, 1 mM $CaCl_2$, 1 mM $MgCl_2$, 20 mM HEPES, pH 7.2). Post injection the worms are incubated at 22°C. After requisite time the injected worms are anesthetized in 40 mM sodium azide in M9 buffer and mounted on a glass slide containing 2% agarose pad. Worms were imaged using Olympus IX83 research inverted microscope (Olympus Corporation of the Americas, Center Valley, PA, USA). The average whole cell intensity in the Alexa 647 channel was plotted as a function of time (*Figure 2—figure supplement 2*)

J774A.1 and THP-1 macrophage cells labeling was carried out using 500 nM $Clensor_{A647}$ using 1X Medium 1 (150 mM NaCl, 5 mM KCl, 1 mM $CaCl_2$, 1 mM $MgCl_2$, 20 mM HEPES, pH 7.2). Cells were pulsed for 30 mins and then chased at 37°C for the indicated time points. The average whole cell intensity in the Alexa 647 channel was plotted as a function of time (*Figure 4—figure supplement 3*).

## Fura-2AM imaging

Cells were loaded with 3 µM Fura-2 AM in HBSS (137 mM NaCl, 5 mM KCl, 1.4 mM $CaCl_2$, 1 mM $MgCl_2$, 0.25 mM $Na_2HPO_4$, 0.44 mM $KH_2PO_4$, 4.2 mM $NaHCO_3$ and 10 mM glucose) at 37°C for 60 min. Post incubation cells were washed with 1X PBS and then imaged in Low $Ca^{2+}$ or Zero $Ca^{2+}$ buffer (145 mM NaCl, 5 mM KCl, 3 mM $MgCl_2$, 10 mM glucose, 1 mM EGTA, and 20 mM HEPES (pH 7.4)). $Ca^{2+}$ concentration in the nominally free $Ca^{2+}$ solution is estimated to be 1–10 µM. With 1 mM EGTA, the free $Ca^{2+}$ concentration is estimated to be <10 nM based on the Maxchelator software (http://maxchelator.stanford.edu/). Florescence was recorded using two different wavelengths (340 and 380 nm) and the ratio ($F_{340}/F_{380}$) was used to calculate changes in intracellular $[Ca^{2+}]$.

## Enzyme activity assays

Enzyme activity assays were performed in J774A.1 cells. For Cathepsin C enzyme activity; we used Gly-Phe $\beta$-naphthylamide as a substrate. Lysosomes of J774A.1 cells were pre-labeled with TMR-dextran (0.5 mg/mL; G) for 1 hr and chased in complete medium for 16 hr at 37°C. The cells were then labeled with 50 nM LysoTracker in complete medium for 30 mins at 37°C. 50 µM NPPB or 200 µM GPN were then added to the cells and incubated for 30 mins at 37°C. The cells then washed and imaged in HBSS buffer containing either NPPB or GPN. The whole cell intensity ratio (G/R) was plotted to quantify the level of LysoTracker labelling of the endosomes. For Cathepsin L and Aryl Sulfatase enzyme activity Magic Red Cathepsin L assay kit (Immunochemistry Technologies) and Lysosomal sulfatase assay kit (Marker Gene) were used. The experiment was performed using the manufacture's protocol. Briefly, cells were incubated with 1X Magic Red Cathepsin L assay probe or 200 µM Lysosomal sulfatase assay probe for 4 hr in complete medium. The cells were then labelled with 10 µM Hoechst stain for 10 mins at 37°C after which the cells were washed and imaged. For low chloride containing dishes; cells were preincubated with 50 µM NPPB before the addition of the enzyme probes. The ratio of enzyme substrate whole cell intensity to that of DAPI was used to quantify enzyme activity.

## Acknowledgements

The authors thank Professors John Kuriyan, Susan Cotman, Drs A H Rahmathullah and D McEwan for critical comments and valuable suggestions. The authors thank the Integrated Light Microscopy facility at the University of Chicago, the *C. elegans* Genetic Center (CGC) for strains, Koushika, S, Glotzer, M, and Ausbel, F for Arhinger Library RNAi clones. Data described in the paper are presented in the main text and the supplementary materials. This work was supported by the National Center for Advancing Translational Sciences of the National Institutes of Health through grant no: UL1 TR000430 and U Chicago startup funds to YK. YK is a Brain Research Foundation Fellow.

## Additional information

### Funding

| Funder | Grant reference number | Author |
|---|---|---|
| Brain Research Foundation | BRF SIA-2016-01 | Yamuna Krishnan |
| National Center for Advancing Translational Sciences | UL1 TR000430 | Yamuna Krishnan |

The funders had no role in study design, data collection and interpretation, or the decision to submit the work for publication.

### Author contributions

KC, Conceptualization, Resources, Data curation, Software, Formal analysis, Validation, Investigation, Methodology, Writing—original draft, Writing—review and editing; KHL, Data curation, Software, Formal analysis, Validation, Investigation, Visualization, Writing—original draft; YK, Conceptualization, Resources, Data curation, Supervision, Funding acquisition, Investigation, Project administration, Writing—original draft, Writing—review and editing

### Author ORCIDs

Kasturi Chakraborty, http://orcid.org/0000-0002-0635-9028
KaHo Leung, http://orcid.org/0000-0002-2998-669X
Yamuna Krishnan, http://orcid.org/0000-0001-5282-8852

## Additional files

### Supplementary files

• Supplementary file 1. Sequences used for *Clensor* and I4$^{cLY}_{A488/A647}$ assemblies. Oligo P, Oligo D1 and Oligo D2 combine to form *Clensor*. Oligo I4$^{cLY}_{A488}$ and I4$^{cLY}_{A647}$ combine to form I4$^{cLY}_{A488/A647}$. The sequences in matching colors are complementary.

• Supplementary file 2. Lysosomal storage disorders investigated in this study, their corresponding human genes and the *C.elegans* homologues.

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
