## [Decision Letter]

[Editors’ note: a previous version of this study was rejected after peer review, but the authors submitted for reconsideration and the paper was accepted for publication. The first decision letter after peer review is shown below.]

Thank you for submitting your work entitled "High lumenal chloride in the lysosome is critical for lysosome function" for consideration by *eLife*. Your article has been favorably evaluated by a Senior Editor and three reviewers, one of whom, Suzanne R Pfeffer (Reviewer #1), is a member of our Board of Reviewing Editors. The following individuals involved in review of your submission have agreed to reveal their identity: Haoxing Xu (Reviewer #2) and Carsten Schultz (Reviewer #3).

Our decision has been reached after consultation between the reviewers. Based on these discussions and the individual reviews below, we regret to inform you that this manuscript will not be considered further for publication in *eLife*. As you will see in the following comments, the reviewers noted a number of strengths in the use of your novel chloride sensor. However, they felt that additional work would be required to bring the work to the level of significance and novelty expected for presentation in *eLife*.

*Reviewer #1:*

Quoting a review from Jentsch (2013), exact measurements of lysosomal [Cl−] have not been reported due to the lack of suitable Cl− sensors for the high [Cl−] at low pH. However, from measurements with cells treated with low Cl− and from model calculations, lysosomal [Cl−]lumen is greater than 80 mM. 2010 data strongly point to a crucial role of Cl− transport in organellar physiology that "goes beyond merely providing the electrical shunt for proton pumping by the V-ATPase." Despite maintaining lysosomal conductance and normal lysosomal pH, Clcn7unc/unc mice showed lysosomal storage disease like mice lacking ClC-7. Although various proteins are known to be regulated by Cl−, the mechanism by which (mainly luminal) Cl− affects membrane traffic and organellar function has remained elusive; the 2010 analysis of Clcn5unc and Clcn7unc mice has suggested that luminal anion concentration is important all along the endosomal-lysosomal pathway.

Here, the authors use a novel, DNA based chloride sensor to try more accurately to determine lysosomal chloride levels. Like the previous work from the Jentsch lab, the findings using *C. elegans* lysosomes indicate a high level of lysosomal chloride. Chloride ion levels are decreased under conditions of lysosomal dysregulation due to mutations in genes that lead to lysosomal storage disorders. Not clear are the mechanisms underlying these observations.

Recently, the lysosome has been shown to represent a significant calcium store, and in worms, CUP-5 is the MCOLN-1 homolog that is important for lysosomal calcium levels and functions. If chloride is not the counterion for protons, it may be important for calcium homeostasis. Given that it is not clear how chloride influences lysosome function in disease states or in normal cells, it would be very important for these authors to clarify the connection between chloride accumulation defects and calcium uptake and regulation. Without this, the findings seem somewhat anecdotal, despite highlighting the recently uncovered importance of chloride ions in lysosome function. No doubt this is an interesting area for further investigation.

*Reviewer #2:*

Cl^-^ transport across endosomal and lysosomal membranes are known to be crucial for lysosomal acidification and physiology, but the underlying mechanisms are poorly understood, largely due to the lack of the methods that can be used to reliably monitor luminal Cl^-^ levels in live cells. The authors previously reported the development of a DNA-based fluorescent reporter (i.e., *Clensor*) for Cl^-^. Compared with protein-based ion-sensing probes, a major advantage for DNA-based probes is their relative insensitivity to pH, which is low in the endosomes and lysosomes. Building on this exciting development in the field, the authors now report the use of *Clensor* in *C. elegans* and mammalian phagocytes, providing evidence that a decrease in lysosomal [Cl^-^] is likely to be a primary pathogenic factor in a number of lysosome storage diseases (LSDs). Overall, the cell biology and imaging experiments in the study were carefully designed, and the results were mostly clean and convincing, and carefully interpreted. Both the use of *Clensor* and the major conclusions of the paper are of substantial interests to researchers working on lysosome biology, anion channels and transporters, membrane transport, organellar channels and transporters, and LSDs. The manuscript can be improved if following points are taken into consideration.

1) Cl^-^ channels are known to exhibit poor anion selectivity. What is the anion selectivity for *Clensor*? When lysosomal [Cl^-^] is reduced in some LSD cells, assuming that lysosome lumen is still electroneutral and iso-osmotic, what are substituted anions? Phosphate is probably an obvious candidate. Therefore, it would be helpful if the authors can show the effects of NO3(-) and PO4(3-) in *Clensor in vitro*.

2) Lysosomal delivery of *Clensor* might be slowed down in LSD cells, compared with wild-type (WT) cells. Could trafficking defects contribute to the observed reduction in lysosomal [Cl^-^]? Given the 40 mM difference in late endosomal vs. lysosomal [Cl^-^], it is plausible that a block in endosome maturation might affect the [Cl^-^] measurement. I would assume that the calibration curves and maximal signals are identical for WT and LSD cells. Is that the case?

*Reviewer #3:*

The work of Krishnan and co-workers describes the use of previously published fluorescent sensors for measuring lysosomal pH and Cl levels. Elegantly, the work was performed in circulating nematode macrophages as well as cultured mammalian cells. This is a very exciting application and the authors observe that a lack of chloride levels correlates with a loss of degrading capacity of the lysosome. The authors follow the hypothesis that the drop-in chloride content is responsible for driving lysosomal function. They tested model conditions for lysosomal storage diseases in *C. elegans* and found indeed reduced chloride levels under such conditions. However, this is not a prove that the chloride ion levels are instrumental. The question remains if chloride levels are the key component in lysosomal dysfunction or a by-stander effect.

The authors argue that CF patients show lysosomal distress symptoms at the molecular level (changes in enzyme activities) similar to Niemann-Pick patients. What the authors do not consider in their argumentation is that CF patients share very little of the NPC phenotype. There is no early neurodegeneration in CF nor a liver accumulation of lipids such as cholesterol or sphingosine. As a result, the conclusions driven by the present study do not hold. Down to the bare bones, the study reveals very interesting data regarding the chloride ion levels in lysosomes but lacks any mechanistic insight. Unfortunately, there is no consideration (or even mentioning) of the calcium levels in the lysosome that have received quite some attention in the regulation of lysosomal function and signaling lately (for instance, see Medina et al. Nature Cell Biology 17, 288-299 (2015)). Chloride might change driving forces for lysosomal im- or export but a direct switching function through changes in protein conformation as is known for calcium ions would be a large surprise.

For publication in *eLife*, I would expect a more profound mechanistic insight of what chloride ions are doing in lysosomes. So far, the authors present highly interesting observations in very relevant cells followed by a conclusion section that is in my opinion not coherent with the observed phenotype (in CF). To make this work suitable for *eLife*, I would expect measuring and/or manipulating calcium levels under conditions where lysosomal chloride levels are high or low, respectively. In addition, the authors could manipulate chloride levels in lysosomes acutely and observe effects spontaneously.

In summary, I suggest to reject this manuscript until the above-mentioned experiment sets have been performed and a mechanistic model is presented.

---

## [Author Response]

[Editors’ note: the author responses to the first round of peer review follow.]

*Reviewer #1:*

*Quoting a review from Jentsch (2013), exact measurements of lysosomal [Cl−] have not been reported due to the lack of suitable Cl− sensors for the high [Cl−] at low pH. However, from measurements with cells treated with low Cl− and from model calculations, lysosomal [Cl−]lumen is greater than 80 mM. 2010 data strongly point to a crucial role of Cl− transport in organellar physiology that "goes beyond merely providing the electrical shunt for proton pumping by the V-ATPase." Despite maintaining lysosomal conductance and normal lysosomal pH, Clcn7unc/unc mice showed lysosomal storage disease like mice lacking ClC-7. Although various proteins are known to be regulated by Cl−, the mechanism by which (mainly luminal) Cl− affects membrane traffic and organellar function has remained elusive; the 2010 analysis of Clcn5unc and Clcn7unc mice has suggested that luminal anion concentration is important all along the endosomal-lysosomal pathway.*

*Here, the authors use a novel, DNA based chloride sensor to try more accurately to determine lysosomal chloride levels. Like the previous work from the Jentsch lab, the findings using C. elegans lysosomes indicate a high level of lysosomal chloride. Chloride ion levels are decreased under conditions of lysosomal dysregulation due to mutations in genes that lead to lysosomal storage disorders. Not clear are the mechanisms underlying these observations.*

*Recently, the lysosome has been shown to represent a significant calcium store, and in worms, CUP-5 is the MCOLN-1 homolog that is important for lysosomal calcium levels and functions. If chloride is not the counterion for protons, it may be important for calcium homeostasis. Given that it is not clear how chloride influences lysosome function in disease states or in normal cells, it would be very important for these authors to clarify the connection between chloride accumulation defects and calcium uptake and regulation. Without this, the findings seem somewhat anecdotal, despite highlighting the recently uncovered importance of chloride ions in lysosome function. No doubt this is an interesting area for further investigation.*

We now include additional experiments that address how low chloride in the lysosome could mechanistically connect to lysosomal storage disease. We pursued two hypotheses. One, suggested by this reviewer, was that low chloride could impede lysosomal Ca^2+^ accumulation, which is known to compromise lysosome function (1–3). We also explored whether chloride could directly impact the proper function of specific lysosomal enzymes (1).

Our studies revealed both mechanisms to be operational. Briefly, selectively inhibiting chloride channels with NPPB resulted in reducing lysosomal Ca^2+^ (Figure 5). Further, the lysosomal enzymes cathepsin C and aryl sulfatase B showed reduced enzymatic activity upon reducing lysosomal Cl^-^ with NPPB (Figure 5).

The results of these experiments are now presented as Figure 5 of the revised manuscript, along with the associated discussion in the subsection “High chloride regulates lysosome function in multiple ways”.

*Reviewer #2:*

*[…] 1) Cl^-^ channels are known to exhibit poor anion selectivity. What is the anion selectivity for Clensor? When lysosomal [Cl^-^] is reduced in some LSD cells, assuming that lysosome lumen is still electroneutral and iso-osmotic, what are substituted anions? Phosphate is probably an obvious candidate. Therefore, it would be helpful if the authors can show the effects of NO3(-) and PO4(3-) in Clensor in vitro*.

We thank the reviewer for this suggestion. Figure 1—figure supplement 3 shows the selectivity of *Clensor* to various anions in the form of their sodium salts. This reveals that the sensitivity of *Clensor* to diverse biologically abundant anions including NO_3_^-^ and PO_4_^3-^ is negligible. This is attributed to the fact that *Clensor's* chloride-sensing module, BAC, undergoes collisional quenching selectively with halide ions because of their well-matched HOMO-LUMO energies (2,3). However, chloride is the most abundant physiological anion (4) hence BAC acts as a chloride sensor in biological systems.

*2) Lysosomal delivery of Clensor might be slowed down in LSD cells, compared with wild-type (WT) cells. Could trafficking defects contribute to the observed reduction in lysosomal [Cl^-^]? Given the 40 mM difference in late endosomal vs. lysosomal [Cl^-^], it is plausible that a block in endosome maturation might affect the [Cl^-^] measurement. I would assume that the calibration curves and maximal signals are identical for WT and LSD cells. Is that the case?*

The reviewer's point is well taken. Before any Cl^-^ measurements were made in any mutant *C. elegans,* we performed colocalization studies of *Clensor* with LMP-1::GFP transgenic worms, where the relevant lysosomal storage disorder gene was knocked down by RNAi. These revealed ~80% colocalization at t = 60 min post injecting *Clensor* in all mutant backgrounds studied here, indicating that trafficking of the probe was not affected. Further, in the context of clh-6 worms, though there is an imbalance in chloride concentrations, lumenal pH of these compartments matches that of wildtype lysosomes suggesting that these compartments are indeed likely to be lysosomes. Several other mutants e.g., ncr-1, aman and manba showed *Clensor* accumulated in grossly misshapen and/or enlarged LMP1::GFP positive compartments, further reinforcing that the compartments are indeed lysosomes swollen due to the accumulated material.

In murine and human macrophages, we pre-labelled lysosomes with TMR-Dextran (0.5 mg/mL) using literature protocols (5), treat cells with U18666A and Conduritol β-epoxide (CBE) to induce Niemann Pick C and Gaucher's cellular phenotype, only then pulse *Clensor* for 30 min, chased for 60 min, and scored for co-localization with TMR-Dextran. Amitriptyline (AH) and NPPB were added 30 min after pulsing and chasing of *Clensor* to lysosomes. We observed that *Clensor* colocalized with lysosomes under each of these conditions, indicating that they do not suffer trafficking defects in these cell culture models of lysosomal storage diseases. This data is now presented as Figure 4—figure supplement 5.

*Reviewer #3:*

*The work of Krishnan and co-workers describes the use of previously published fluorescent sensors for measuring lysosomal pH and Cl levels. Elegantly, the work was performed in circulating nematode macrophages as well as cultured mammalian cells. This is a very exciting application and the authors observe that a lack of chloride levels correlates with a loss of degrading capacity of the lysosome. The authors follow the hypothesis that the drop-in chloride content is responsible for driving lysosomal function. They tested model conditions for lysosomal storage diseases in C. elegans and found indeed reduced chloride levels under such conditions. However, this is not a prove that the chloride ion levels are instrumental. The question remains if chloride levels are the key component in lysosomal dysfunction or a by-stander effect.*

Please see response to reviewer #1, point #1. In a nutshell, this is not a by-stander effect.

*The authors argue that CF patients show lysosomal distress symptoms at the molecular level (changes in enzyme activities) similar to Niemann-Pick patients. What the authors do not consider in their argumentation is that CF patients share very little of the NPC phenotype. There is no early neurodegeneration in CF nor a liver accumulation of lipids such as cholesterol or sphingosine. As a result, the conclusions driven by the present study do not hold.*

Actually, we never implied any connection between Niemann Pick C with CF. We only cited published literature that correlated reduced arylsulfatase activity and acid sphingomyelinase activity in cells derived from patients with CF (12,13). These enzymes are defective in mucopolysaccharidoses VI and Niemann Pick A/B diseases(14). These are distinct diseases from Niemann Pick C disease.

However, given the extensive new data that we added, we have dropped mention of CF as it can prove distracting to the reader.

*Down to the bare bones, the study reveals very interesting data regarding the chloride ion levels in lysosomes but lacks any mechanistic insight. Unfortunately, there is no consideration (or even mentioning) of the calcium levels in the lysosome that have received quite some attention in the regulation of lysosomal function and signaling lately (for instance, see Medina et al. Nature Cell Biology 17, 288-299 (2015)). Chloride might change driving forces for lysosomal im- or export but a direct switching function through changes in protein conformation as is known for calcium ions would be a large surprise.*

Please see response to reviewer #1, point #1. Briefly, lysosomal Ca^2+^ is indeed depleted when lysosomal chloride is reduced. However, the functions of specific lysosomal enzymes such as cathepsin C and aryl sulfatase B, is also affected. There is *in vitro* evidence that high Cl^-^ is necessary for the activity of these enzymes (15–19). Data to this effect is now presented as Figure 5 in the revised main manuscript.

*For publication in eLife, I would expect a more profound mechanistic insight of what chloride ions are doing in lysosomes. So far, the authors present highly interesting observations in very relevant cells followed by a conclusion section that is in my opinion not coherent with the observed phenotype (in CF). To make this work suitable for eLife, I would expect measuring and/or manipulating calcium levels under conditions where lysosomal chloride levels are high or low, respectively. In addition, the authors could manipulate chloride levels in lysosomes acutely and observe effects spontaneously.*

We have addressed this comment. Please see response to reviewer #1, point #1.